# FLASHATTENTION: Fast and Memory-Efficient Exact Attention with IO-Awareness

**Tri Dao**[†], **Daniel Y. Fu** [†], **Stefano Ermon** [†], **Atri Rudra** [‡], **Christopher Ré** [†]

[†] Department of Computer Science, Stanford University

[‡] Department of Computer Science and Engineering, University at Buffalo, SUNY

{trid,danfu}@stanford.edu, ermon@stanford.edu, atri@buffalo.edu, chrismre@cs.stanford.edu

## Abstract

Transformers are slow and memory-hungry on long sequences, since the time and memory complexity of self-attention are quadratic in sequence length. Approximate attention methods have attempted to address this problem by trading off model quality to reduce the compute complexity, but often do not achieve wall-clock speedup. We argue that a missing principle is making attention algorithms *IO-aware*— accounting for reads and writes between levels of GPU memory. We propose FLASHATTENTION, an IO-aware exact attention algorithm that uses tiling to reduce the number of memory reads/writes between GPU high bandwidth memory (HBM) and GPU on-chip SRAM. We analyze the IO complexity of FLASHATTENTION, showing that it requires fewer HBM accesses than standard attention, and is optimal for a range of SRAM sizes. We also extend FLASHATTENTION to block-sparse attention, yielding an approximate attention algorithm that is faster than any existing approximate attention method. FLASHATTENTION trains Transformers faster than existing baselines: 15% end-to-end wall-clock speedup on BERT-large (seq. length 512) compared to the MLPerf 1.1 training speed record, 3× speedup on GPT-2 (seq. length 1K), and 2.4× speedup on long-range arena (seq. length 1K-4K). FLASHATTENTION and block-sparse FLASHATTENTION enable longer context in Transformers, yielding higher quality models (0.7 better perplexity on GPT-2 and 6.4 points of lift on long-document classification) and entirely new capabilities: the first Transformers to achieve better-than-chance performance on the Path-X challenge (seq. length 16K, 61.4% accuracy) and Path-256 (seq. length 64K, 63.1% accuracy).

## 1 Introduction

Transformer models [86] have emerged as the most widely used architecture in applications such as natural language processing and image classification. Transformers have grown larger [5] and deeper [87], but equipping them with longer context remains difficult [83], since the self-attention module at their heart has time and memory complexity quadratic in sequence length. An important question is whether making attention faster and more memory-efficient can help Transformer models address their runtime and memory challenges for long sequences.

Many approximate attention methods have aimed to reduce the compute and memory requirements of attention. These methods range from sparse-approximation [53, 77] to low-rank approximation [13, 52, 88], and their combinations [3, 9, 96]. Although these methods reduce the compute requirements to linear or near-linear in sequence length, many of them do not display wall-clock speedup against standard attention and have not gained wide adoption. One main reason is that they focus on FLOP reduction (which may not correlate with wall-clock speed) and tend to ignore overheads from memory access (IO).

In this paper, we argue that a missing principle is making attention algorithms *IO-aware* [1]—that is, carefully accounting for reads and writes to different levels of fast and slow memory (e.g., between fast GPU on-chip SRAM and relatively slow GPU high bandwidth memory, or HBM [47], Figure 1

36th Conference on Neural Information Processing Systems (NeurIPS 2022).

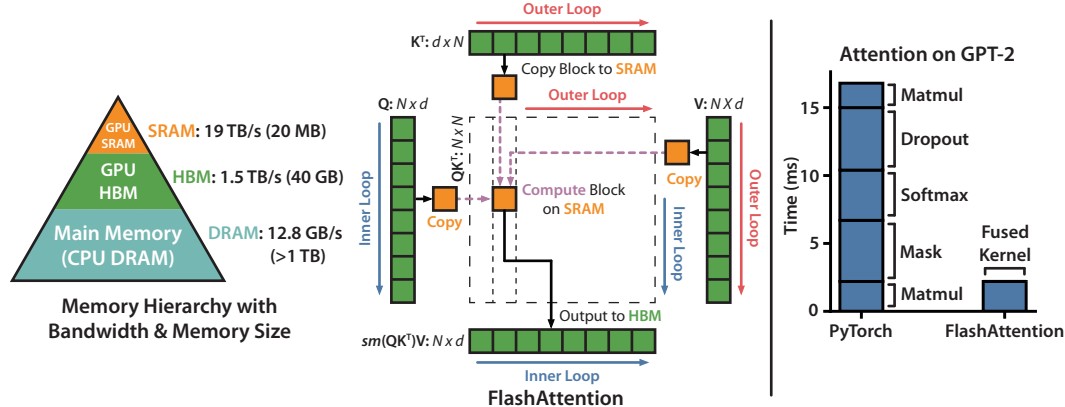

Figure 1: **Left:** FLASHATTENTION uses tiling to prevent materialization of the large $N \times N$ attention matrix (dotted box) on (relatively) slow GPU HBM. In the outer loop (red arrows), FLASHATTENTION loops through blocks of the **K** and **V** matrices and loads them to fast on-chip SRAM. In each block, FLASHATTENTION loops over blocks of **Q** matrix (blue arrows), loading them to SRAM, and writing the output of the attention computation back to HBM. **Right:** Speedup over the PyTorch implementation of attention on GPT-2. FLASHATTENTION does not read and write the large $N \times N$ attention matrix to HBM, resulting in an 7.6× speedup on the attention computation.

left). On modern GPUs, compute speed has out-paced memory speed [64–66], and most operations in Transformers are bottlenecked by memory accesses [45]. IO-aware algorithms have been critical for similar memory-bound operations, when reading and writing data can account for a large portion of the runtime—such as database joins [74], image processing [73], numerical linear algebra [4], and more [42, 89]. However, common Python interfaces to deep learning such as PyTorch and Tensorflow do not allow fine-grained control of memory access.

We propose FLASHATTENTION, a new attention algorithm that computes exact attention with far fewer memory accesses. Our main goal is to avoid reading and writing the attention matrix to and from HBM. This requires (i) computing the softmax reduction without access to the whole input (ii) not storing the large intermediate attention matrix for the backward pass. We apply two well-established techniques to address these challenges. (i) We restructure the attention computation to split the input into blocks and make several passes over input blocks, thus incrementally performing the softmax reduction (also known as **tiling**). (ii) We store the softmax normalization factor from the forward pass to quickly **recompute** attention on-chip in the backward pass, which is faster than the standard approach of reading the intermediate attention matrix from HBM. We implement FLASHATTENTION in CUDA to achieve fine-grained control over memory access and fuse all the attention operations into one GPU kernel. Even with the increased FLOPs due to recomputation, our algorithm both **runs faster** (up to 7.6x on GPT-2 [70], Figure 1 right) and **uses less memory**—linear in sequence length—than standard attention, thanks to the massively reduced amount of HBM access.

We analyze the IO complexity [1] of FLASHATTENTION, proving that it requires $O(N^2 d^2 M^{-1})$ HBM accesses where $d$ is the head dimension and $M$ is the size of SRAM, as compared to $\Omega(Nd+N^2)$ of standard attention. For typical values of $d$ and $M$, FLASHATTENTION requires many times fewer HBM accesses compared to standard attention (up to 9× fewer, as shown in Fig. 2). Moreover, we provide a lower bound, showing that no exact attention algorithm can asymptotically improve on the number of HBM accesses over all SRAM sizes.

We also show that FLASHATTENTION can serve as a useful primitive for realizing the potential of approximate attention algorithms by overcoming their issues with memory access overhead. As a proof of concept, we implement block-sparse FLASHATTENTION, a sparse attention algorithm that is 2-4× faster than even FLASHATTENTION, scaling up to sequence length of 64k. We prove that block-sparse FLASHATTENTION has better IO complexity than FLASHATTENTION by a factor proportional to the sparsity ratio. We discuss further extensions to other operations (attention on multi-GPU, kernel regression, block-sparse matrix multiply) in Section 5. We open-source FLASHATTENTION to make it easier to build on this primitive[1].

---

[1] FLASHATTENTION code is available at `https://github.com/HazyResearch/flash-attention`

We empirically validate that FLASHATTENTION speeds up model training and improves model quality by modeling longer context. We also benchmark the runtime and memory footprint of FLASHAT-TENTION and block-sparse FLASHATTENTION compared to prior attention implementations.

- **Faster Model Training.** FLASHATTENTION trains Transformer models faster in wall-clock time. We train BERT-large (seq. length 512) 15% faster than the training speed record in MLPerf 1.1 [60], GPT2 (seq. length 1K) 3× faster than baseline implementations from HuggingFace [91] and Megatron-LM [80], and long-range arena (seq. length 1K-4K) 2.4× faster than baselines.

- **Higher Quality Models.** FLASHATTENTION scales Transformers to longer sequences, which improves their quality and enables new capabilities. We observe a 0.7 improvement in perplexity on GPT-2 and 6.4 points of lift from modeling longer sequences on long-document classification [14]. FLASHATTENTION enables the first Transformer that can achieve better-than-chance performance on the Path-X [83] challenge, solely from using a longer sequence length (16K). Block-sparse FLASHATTENTION enables a Transformer to scale to even longer sequences (64K), resulting in the first model that can achieve better-than-chance performance on Path-256.

- **Benchmarking Attention.** FLASHATTENTION is up to 3× faster than the standard attention implementation across common sequence lengths from 128 to 2K and scales up to 64K. Up to sequence length of 512, FLASHATTENTION is both faster and more memory-efficient than any existing attention method, whereas for sequence length beyond 1K, some approximate attention methods (e.g., Linformer) start to become faster. On the other hand, block-sparse FLASHATTENTION is faster than all existing approximate attention methods that we know of.

## 2 Background

We provide some background on the performance characteristics of common deep learning operations on modern hardware (GPUs). We also describe the standard implementation of attention.

### 2.1 Hardware Performance

We focus here on GPUs. Performance on other hardware accelerators are similar [48, 50].

**GPU Memory Hierarchy.** The GPU memory hierarchy (Fig. 1 left) comprises multiple forms of memory of different sizes and speeds, with smaller memory being faster. As an example, the A100 GPU has 40-80GB of high bandwidth memory (HBM) with bandwidth 1.5-2.0TB/s and 192KB of on-chip SRAM per each of 108 streaming multiprocessors with bandwidth estimated around 19TB/s [46, 47]. The on-chip SRAM is an order of magnitude faster than HBM but many orders of magnitude smaller in size. As compute has gotten faster relative to memory speed [64–66], operations are increasingly bottlenecked by memory (HBM) accesses. Thus exploiting fast SRAM becomes more important.

**Execution Model.** GPUs have a massive number of threads to execute an operation (called a kernel). Each kernel loads inputs from HBM to registers and SRAM, computes, then writes outputs to HBM.

**Performance characteristics.** Depending on the balance of computation and memory accesses, operations can be classified as either compute-bound or memory-bound. This is commonly measured by the *arithmetic intensity* [89], which is the number of arithmetic operations per byte of memory access.

1. Compute-bound: the time taken by the operation is determined by how many arithmetic operations there are, while time accessing HBM is much smaller. Typical examples are matrix multiply with large inner dimension, and convolution with large number of channels.

2. Memory-bound: the time taken by the operation is determined by the number of memory accesses, while time spent in computation is much smaller. Examples include most other operations: elementwise (e.g., activation, dropout), and reduction (e.g., sum, softmax, batch norm, layer norm).

**Kernel fusion.** The most common approach to accelerate memory-bound operations is kernel fusion: if there are multiple operations applied to the same input, the input can be loaded once from HBM, instead of multiple times for each operation. Compilers can automatically fuse many elementwise operations [55, 68, 78]. However, in the context of model training, the intermediate values still need to be written to HBM to save for the backward pass, reducing the effectiveness of naive kernel fusion.

### 2.2 Standard Attention Implementation

Given input sequences $\mathbf{Q}, \mathbf{K}, \mathbf{V} \in \mathbb{R}^{N \times d}$ where $N$ is the sequence length and $d$ is the head dimension, we want to compute the attention output $\mathbf{O} \in \mathbb{R}^{N \times d}$:

$$\mathbf{S} = \mathbf{Q}\mathbf{K}^\top \in \mathbb{R}^{N \times N}, \quad \mathbf{P} = \text{softmax}(\mathbf{S}) \in \mathbb{R}^{N \times N}, \quad \mathbf{O} = \mathbf{P}\mathbf{V} \in \mathbb{R}^{N \times d},$$

where softmax is applied row-wise.

Standard attention implementations materialize the matrices $\mathbf{S}$ and $\mathbf{P}$ to HBM, which takes $O(N^2)$ memory. Often $N \gg d$ (e.g., for GPT2, $N = 1024$ and $d = 64$). We describe the standard attention implementation in Algorithm 0. As some or most of the operations are memory-bound (e.g., softmax), the large number of memory accesses translates to slow wall-clock time.

This problem is exacerbated by other elementwise operations applied to the attention matrix, such as masking applied to $\mathbf{S}$ or dropout applied to $\mathbf{P}$. As a result, there have been many attempts to fuse several elementwise operations, such as fusing masking with softmax [80].

In Section 3.2, we will show that the standard attention implementation performs HBM accesses quadratic in the sequence length $N$. We also compare the number of FLOPs and number of HBM accesses of standard attention and of our method (FLASHATTENTION).

---

**Algorithm 0** Standard Attention Implementation

---

**Require:** Matrices $\mathbf{Q}, \mathbf{K}, \mathbf{V} \in \mathbb{R}^{N \times d}$ in HBM.
 1: Load $\mathbf{Q}, \mathbf{K}$ by blocks from HBM, compute $\mathbf{S} = \mathbf{Q}\mathbf{K}^\top$, write $\mathbf{S}$ to HBM.
 2: Read $\mathbf{S}$ from HBM, compute $\mathbf{P} = \text{softmax}(\mathbf{S})$, write $\mathbf{P}$ to HBM.
 3: Load $\mathbf{P}$ and $\mathbf{V}$ by blocks from HBM, compute $\mathbf{O} = \mathbf{P}\mathbf{V}$, write $\mathbf{O}$ to HBM.
 4: Return $\mathbf{O}$.

---

## 3 FLASHATTENTION: Algorithm, Analysis, and Extensions

We show how to compute exact attention with fewer HBM reads/writes and without storing large intermediate matrices for the backward pass. This yields an attention algorithm that is both memory efficient and faster in wall-clock time. We analyze its IO complexity, showing that our method requires much fewer HBM accesses compared to standard attention. We further show that FLASHATTENTION can serve as a useful primitive by extending it to handle block-sparse attention.

We focus here on the forward pass for ease of exposition; Appendix B contains details for the backward.

### 3.1 An Efficient Attention Algorithm With Tiling and Recomputation

Given the inputs $\mathbf{Q}, \mathbf{K}, \mathbf{V} \in \mathbb{R}^{N \times d}$ in HBM, we aim to compute the attention output $\mathbf{O} \in \mathbb{R}^{N \times d}$ and write it to HBM. Our goal is to reduce the amount of HBM accesses (to sub-quadratic in $N$).

We apply two established techniques (tiling, recomputation) to overcome the technical challenge of computing exact attention in sub-quadratic HBM accesses. We describe this in Algorithm 1. The main idea is that we split the inputs $\mathbf{Q}, \mathbf{K}, \mathbf{V}$ into blocks, load them from slow HBM to fast SRAM, then compute the attention output with respect to those blocks. By scaling the output of each block by the right normalization factor before adding them up, we get the correct result at the end.

**Tiling.** We compute attention by blocks. Softmax couples columns of $\mathbf{K}$, so we decompose the large softmax with scaling [53, 62, 69]. For numerical stability, the softmax of vector $x \in \mathbb{R}^B$ is computed:

$$m(x) := \max_i x_i, \quad f(x) := \begin{bmatrix} e^{x_1 - m(x)} & \dots & e^{x_B - m(x)} \end{bmatrix}, \quad \ell(x) := \sum_i f(x)_i, \quad \text{softmax}(x) := \frac{f(x)}{\ell(x)}.$$

For vectors $x^{(1)}, x^{(2)} \in \mathbb{R}^B$, we can decompose the softmax of the concatenated $x = \begin{bmatrix} x^{(1)} & x^{(2)} \end{bmatrix} \in \mathbb{R}^{2B}$ as:

$$m(x) = m(\begin{bmatrix} x^{(1)} & x^{(2)} \end{bmatrix}) = \max(m(x^{(1)}), m(x^{(2)})), \quad f(x) = \begin{bmatrix} e^{m(x^{(1)}) - m(x)} f(x^{(1)}) & e^{m(x^{(2)}) - m(x)} f(x^{(2)}) \end{bmatrix},$$

$$\ell(x) = \ell(\begin{bmatrix} x^{(1)} & x^{(2)} \end{bmatrix}) = e^{m(x^{(1)}) - m(x)} \ell(x^{(1)}) + e^{m(x^{(2)}) - m(x)} \ell(x^{(2)}), \quad \text{softmax}(x) = \frac{f(x)}{\ell(x)}.$$

Therefore if we keep track of some extra statistics $(m(x), \ell(x))$, we can compute softmax one block at a time.[2] We thus split the inputs $\mathbf{Q}, \mathbf{K}, \mathbf{V}$ into blocks (Algorithm 1 line 1), compute the softmax values along with extra statistics (Algorithm 1 line 1), and combine the results (Algorithm 1 line 1).

**Recomputation.** One of our goals is to not store $O(N^2)$ intermediate values for the backward pass. The backward pass typically requires the matrices $\mathbf{S}, \mathbf{P} \in \mathbb{R}^{N \times N}$ to compute the gradients with respect to $\mathbf{Q}, \mathbf{K}, \mathbf{V}$. However, by storing the output $\mathbf{O}$ and the softmax normalization statistics $(m, \ell)$, we can recompute the attention matrix $\mathbf{S}$ and $\mathbf{P}$ easily in the backward pass from blocks of $\mathbf{Q}, \mathbf{K}, \mathbf{V}$ in SRAM. This can be seen as a form of selective gradient checkpointing [10, 36]. While gradient

---

[2]This style of aggregation is called *algebraic aggregation* [35].

checkpointing has been suggested to reduce the maximum amount of memory required [69], all implementations (that we know off) have to trade speed for memory. In contrast, even with more FLOPs, our recomputation speeds up the backward pass due to reduced HBM accesses (Fig. 2). The full backward pass description is in Appendix B.

**Implementation details: Kernel fusion.** Tiling enables us to implement our algorithm in one CUDA kernel, loading input from HBM, performing all the computation steps (matrix multiply, softmax, optionally masking and dropout, matrix multiply), then write the result back to HBM (masking and dropout in Appendix B). This avoids repeatedly reading and writing of inputs and outputs from and to HBM.

---

**Algorithm 1** FLASHATTENTION

---

**Require:** Matrices $\mathbf{Q},\mathbf{K},\mathbf{V}\in\mathbb{R}^{N\times d}$ in HBM, on-chip SRAM of size $M$.
1: Set block sizes $B_c=\left\lceil\frac{M}{4d}\right\rceil,B_r=\min\left(\left\lceil\frac{M}{4d}\right\rceil,d\right)$.
2: Initialize $\mathbf{O}=(0)_{N\times d}\in\mathbb{R}^{N\times d},\ell=(0)_N\in\mathbb{R}^N,m=(-\infty)_N\in\mathbb{R}^N$ in HBM.
3: Divide $\mathbf{Q}$ into $T_r=\left\lceil\frac{N}{B_r}\right\rceil$ blocks $\mathbf{Q}_1,...,\mathbf{Q}_{T_r}$ of size $B_r\times d$ each, and divide $\mathbf{K},\mathbf{V}$ in to $T_c=\left\lceil\frac{N}{B_c}\right\rceil$ blocks $\mathbf{K}_1,...,\mathbf{K}_{T_c}$ and $\mathbf{V}_1,...,\mathbf{V}_{T_c}$, of size $B_c\times d$ each.
4: Divide $\mathbf{O}$ into $T_r$ blocks $\mathbf{O}_i,...,\mathbf{O}_{T_r}$ of size $B_r\times d$ each, divide $\ell$ into $T_r$ blocks $\ell_i,...,\ell_{T_r}$ of size $B_r$ each, divide $m$ into $T_r$ blocks $m_1,...,m_{T_r}$ of size $B_r$ each.
5: **for** $1\leq j\leq T_c$ **do**
6:     Load $\mathbf{K}_j,\mathbf{V}_j$ from HBM to on-chip SRAM.
7:     **for** $1\leq i\leq T_r$ **do**
8:         Load $\mathbf{Q}_i,\mathbf{O}_i,\ell_i,m_i$ from HBM to on-chip SRAM.
9:         On chip, compute $\mathbf{S}_{ij}=\mathbf{Q}_i\mathbf{K}_j^T\in\mathbb{R}^{B_r\times B_c}$.
10:        On chip, compute $\tilde{m}_{ij}=\text{rowmax}(\mathbf{S}_{ij})\in\mathbb{R}^{B_r}$, $\tilde{\mathbf{P}}_{ij}=\exp(\mathbf{S}_{ij}-\tilde{m}_{ij})\in\mathbb{R}^{B_r\times B_c}$ (pointwise), $\tilde{\ell}_{ij}=\text{rowsum}(\tilde{\mathbf{P}}_{ij})\in\mathbb{R}^{B_r}$.
11:        On chip, compute $m_i^{\text{new}}=\max(m_i,\tilde{m}_{ij})\in\mathbb{R}^{B_r}$, $\ell_i^{\text{new}}=e^{m_i-m_i^{\text{new}}}\ell_i+e^{\tilde{m}_{ij}-m_i^{\text{new}}}\tilde{\ell}_{ij}\in\mathbb{R}^{B_r}$.
12:        Write $\mathbf{O}_i\leftarrow\text{diag}(\ell_i^{\text{new}})^{-1}(\text{diag}(\ell_i)e^{m_i-m_i^{\text{new}}}\mathbf{O}_i+e^{\tilde{m}_{ij}-m_i^{\text{new}}}\tilde{\mathbf{P}}_{ij}\mathbf{V}_j)$ to HBM.
13:        Write $\ell_i\leftarrow\ell_i^{\text{new}}$, $m_i\leftarrow m_i^{\text{new}}$ to HBM.
14:     **end for**
15: **end for**
16: Return $\mathbf{O}$.

---

We show FLASHATTENTION's correctness, runtime, and memory requirement (proof in Appendix C).

**Theorem 1.** *Algorithm 1 returns* $\mathbf{O}=\text{softmax}(\mathbf{Q}\mathbf{K}^\top)\mathbf{V}$ *with* $O(N^2d)$ *FLOPs and requires* $O(N)$ *additional memory beyond inputs and output.*

### 3.2 Analysis: IO Complexity of FLASHATTENTION

We analyze the IO complexity of FLASHATTENTION, showing significant reduction in HBM accesses compared to standard attention. We also provide a lower bound, proving that no exact attention algorithm can asymptotically improve on HBM accesses over all SRAM sizes. Proofs are in Appendix C.

**Theorem 2.** *Let* $N$ *be the sequence length,* $d$ *be the head dimension, and* $M$ *be size of SRAM with* $d\leq M\leq Nd$*. Standard attention (Algorithm 0) requires* $\Theta(Nd+N^2)$ *HBM accesses, while* FLASHATTENTION *(Algorithm 1) requires* $\Theta(N^2d^2M^{-1})$ *HBM accesses.*

For typical values of $d$ (64-128) and $M$ (around 100KB), $d^2$ is many times smaller than $M$, and thus FLASHATTENTION requires many times fewer HBM accesses than standard implementation. This leads to both faster execution and lower memory footprint, which we validate in Section 4.3.

The main idea of the proof is that given the SRAM size of $M$, we can load blocks of $\mathbf{K},\mathbf{V}$ of size $\Theta(M)$ each (Algorithm 1 line 1). For each block of $\mathbf{K}$ and $\mathbf{V}$, we iterate over all blocks of $\mathbf{Q}$ (Algorithm 1 line 1) to compute the intermediate values, resulting in $\Theta(NdM^{-1})$ passes over $\mathbf{Q}$. Each pass loads $\Theta(Nd)$ elements, which amounts to $\Theta(N^2d^2M^{-1})$ HBM accesses. We similarly prove that the backward pass of standard attention requires $\Theta(Nd+N^2)$ HBM accesses while the backward pass of FLASHATTENTION requires $\Theta(N^2d^2M^{-1})$ HBM accesses (Appendix B).

We prove a lower-bound: one cannot asymptotically improve on the number of HBM accesses for all values of $M$ (the SRAM size) when computing exact attention.

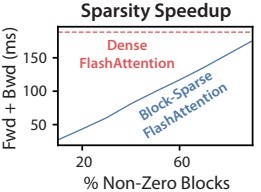

| Attention | Standard | FLASHATTENTION |
|---|---|---|
| GFLOPs | 66.6 | 75.2 |
| HBM R/W (GB) | 35.3 | 4.4 |
| Runtime (ms) | 35.1 | 11.7 |

Figure 2: **Left**: Forward + backward runtime of standard attention and FLASHATTENTION for seq. length 1024, head dim. 64, 16 heads, batch size 64, key-padding mask and no dropout on A100 GPU. HBM access is one of the primary factors affecting runtime. **Middle**: Forward runtime of FLASHATTENTION (seq. length 1024, head dim. 64, 16 heads, batch size 64) on A100 GPU. Fewer HBM accesses result in faster runtime, up to a point. **Right**: The runtime (for seq. length 4K) of block-sparse FLASHATTENTION is faster than FLASHATTENTION by a factor proportional to the sparsity.

**Proposition 3.** *Let N be the sequence length, d be the head dimension, and M be size of SRAM with $d \le M \le Nd$. There does not exist an algorithm to compute exact attention with $o(N^2 d^2 M^{-1})$ HBM accesses for all M in the range $[d, Nd]$.*

The proof relies on the fact that for $M = \Theta(Nd)$ any algorithm must perform $\Omega(N^2 d^2 M^{-1}) = \Omega(Nd)$ HBM accesses. This type of lower bound over a subrange of $M$ is common in the streaming algorithms literature [92]. We leave proving parameterized complexity [29] lower bounds in terms of $M$ as exciting future work.

We validate that the number of HBM accesses is the main determining factor of attention run-time. In Fig. 2 (left), we see that even though FLASHATTENTION has higher FLOP count compared to standard attention (due to recomputation in the backward pass), it has much fewer HBM accesses, resulting in much faster runtime. In Fig. 2 (middle), we vary the block size $B_c$ of FLASHATTENTION, which results in different amounts of HBM accesses, and measure the runtime of the forward pass. As block size increases, the number of HBM accesses decreases (as we make fewer passes over the input), and runtime decreases. For large enough block size (beyond 256), the runtime is then bottlenecked by other factors (e.g., arithmetic operations). Moreover, larger block size will not fit into the small SRAM size.

### 3.3 Extension: Block-Sparse FLASHATTENTION

We extend FLASHATTENTION to approximate attention: we propose block-sparse FLASHATTENTION, whose IO complexity is smaller than FLASHATTENTION by a factor proportional to the sparsity.

Given inputs $\mathbf{Q}, \mathbf{K}, \mathbf{V} \in \mathbb{R}^{N \times d}$ and a mask matrix $\tilde{\mathbf{M}} \in \{0,1\}^{N \times N}$, we want to compute:
$$\mathbf{S} = \mathbf{Q}\mathbf{K}^\top \in \mathbb{R}^{N \times N}, \quad \mathbf{P} = \text{softmax}(\mathbf{S} \odot \mathbb{1}_{\tilde{\mathbf{M}}}) \in \mathbb{R}^{N \times N}, \quad \mathbf{O} = \mathbf{P}\mathbf{V} \in \mathbb{R}^{N \times d},$$
where $(\mathbf{S} \odot \mathbb{1}_{\tilde{\mathbf{M}}})_{kl} = \mathbf{S}_{kl}$ if $\tilde{\mathbf{M}}_{kl} = 1$ and $-\infty$ if $\mathbf{M}_{kl} = 0$. We require $\tilde{\mathbf{M}}$ to have block form: for some block sizes $B_r, B_c$, for all $k, l$, $\tilde{\mathbf{M}}_{k,l} = \mathbf{M}_{ij}$ with $i = \lfloor k/B_r \rfloor, j = \lfloor l/B_c \rfloor$ for some $\mathbf{M} \in \{0,1\}^{N/B_r \times N/B_c}$.

Given a predefined block sparsity mask $\mathbf{M} \in \{0,1\}^{N/B_r \times N/B_c}$ we can easily adapt Algorithm 1 to only compute the nonzero blocks of the attention matrix. The algorithm is identical to Algorithm 1, except we skip zero blocks. We reproduce the algorithm description in Algorithm 5 in Appendix B.

We also analyze the IO complexity of block-sparse FLASHATTENTION.

**Proposition 4.** *Let N be the sequence length, d be the head dimension, and M be size of SRAM with $d \le M \le Nd$. Block-sparse FLASHATTENTION (Algorithm 5) requires $\Theta(Nd + N^2 d^2 M^{-1} s)$ HBM accesses where s is the fraction of nonzero blocks in the block-sparsity mask.*

We see that applying block-sparsity yields a direct improvement by the sparsity to the larger term in the IO complexity. For large sequence lengths $N$, $s$ is often set to $N^{-1/2}$ [12] or $N^{-1} \log N$ [3, 18, 96], resulting in $\Theta(N\sqrt{N})$ or $\Theta(N \log N)$ IO complexity. For downstream experiments, we use the fixed butterfly sparsity pattern [18], which has been shown to be able to approximate arbitrary sparsity [17].

In Fig. 2 (right), we validate that as the sparsity increases, the runtime of block-sparse FLASHATTENTION improves proportionally. On the LRA benchmark, block-sparse FLASHATTENTION achieves 2.8× speedup, while performing on par with standard attention (Section 4).

## 4 Experiments

We evaluate the impact of using FLASHATTENTION to train Transformer models. We validate two claims about training time and model accuracy, and report attention runtime and memory benchmarks.

- **Training Speed.** FLASHATTENTION outperforms the MLPerf 1.1 [60] speed record for BERT by 15%, and speeds up GPT-2 up to 3× over HuggingFace [91] and 1.8× over Megatron [80] over standard Transformers. FLASHATTENTION speeds up the long-range arena (LRA) benchmark 2.4×.
- **Quality.** FLASHATTENTION scales Transformers to longer sequences, yielding higher quality. FLASHATTENTION trains GPT-2 with context length 4K faster than Megatron trains GPT-2 with context length 1K, while achieving 0.7 better perplexity. Modeling longer sequences yields 6.4 points of lift on two long-document classification tasks. Finally, FLASHATTENTION yields the **first Transformer** that can achieve better-than-random performance on the challenging Path-X task (sequence length 16K), and block-sparse FLASHATTENTION yields the **first sequence model** that we know of that can achieve better-than-random performance on Path-256 (sequence length 64K).
- **Benchmarking Attention.** We measure the runtime and memory performance of FLASHATTENTION and block-sparse FLASHATTENTION based on sequence length. We confirm that the memory footprint of FLASHATTENTION scales linearly with seq. length and is up to 3× faster than standard attention for common seq. lengths (up to 2K). We confirm that runtime of block-sparse FLASHATTENTION scales linearly in seq. length and is faster than all existing approximate attention baselines.

Additional experiment details are in Appendix E.

## 4.1 Faster Models with FLASHATTENTION

**BERT.** FLASHATTENTION yields the fastest single-node BERT training speed that we know of. We train a BERT-large [24] model with FLASHATTENTION on Wikipedia. Table 1 compares our training time to the implementation from Nvidia that set the training speed record for MLPerf 1.1 [60, 63]. Our implementation is 15% faster.

Table 1: Training time of BERT-large, starting from the same initialization provided by the MLPerf benchmark, to reach the target accuracy of 72.0% on masked language modeling. Averaged over 10 runs on 8×A100 GPUs.

| BERT Implementation | Training time (minutes) |
|---|---|
| Nvidia MLPerf 1.1 [63] | 20.0 ± 1.5 |
| FLASHATTENTION (ours) | **17.4** ± 1.4 |

**GPT-2.** FLASHATTENTION yields faster training times for GPT-2 [70] on the large OpenWebtext dataset [34] than the widely used HuggingFace [91] and Megatron-LM [80] implementations. Table 2 shows up to 3× end-to-end speedup compared to Huggingface and 1.7× speedup compared to Megatron-LM. FLASHATTENTION achieves the same perplexity as the other two implementations, as we do not change the model definition. Appendix E includes plots of the validation perplexity throughout training, confirming that FLASHATTENTION is as numerically stable as the baselines and produces the same training / validation curves.

Table 2: GPT-2 small and medium using FLASHATTENTION achieve up to 3× speed up compared to Huggingface implementation and up to 1.7× compared to Megatron-LM. Training time reported on 8×A100s GPUs.

| Model implementations | OpenWebText (ppl) | Training time (speedup) |
|---|---|---|
| GPT-2 small - Huggingface [91] | 18.2 | 9.5 days (1.0×) |
| GPT-2 small - Megatron-LM [80] | 18.2 | 4.7 days (2.0×) |
| GPT-2 small - FLASHATTENTION | 18.2 | **2.7 days (3.5×)** |
| GPT-2 medium - Huggingface [91] | 14.2 | 21.0 days (1.0×) |
| GPT-2 medium - Megatron-LM [80] | 14.2 | 11.5 days (1.8×) |
| GPT-2 medium - FLASHATTENTION | 14.2 | **6.9 days (3.0×)** |

**Long-range Arena.** We compare vanilla Transformer (with either standard implementation or FLASHATTENTION) on the long-range arena (LRA [83]) benchmark. We measure accuracy, throughput, and training time of all models. Each task has a different sequence length varying between 1024 and 4096. We follow the implementation and experimental setting in Tay et al. [83] and Xiong et al. [94].[3] Table 3 shows that FLASHATTENTION achieves up 2.4× speed-up compared to standard attention. Block-sparse FLASHATTENTION is faster than all of the approximate attention methods that we have tested.

## 4.2 Better Models with Longer Sequences

**Language Modeling with Long Context.** The runtime and memory-efficiency of FLASHATTENTION allow us to increase the context length of GPT-2 by 4× while still running faster than the optimized implementation from Megatron-LM. Table 4 shows that that GPT-2 with

---

[3]LRA accuracy results are known to be highly dependent on the tuning procedure [94]. Our reproduced baselines perform better than as reported in the original comparison [83].

Table 3: The performance of standard attention, FLASHATTENTION, block-sparse FLASHATTENTION, and approximate attention baselines on the Long-Range-Arena benchmarks.

| Models | ListOps | Text | Retrieval | Image | Pathfinder | Avg | Speedup |
|---|---|---|---|---|---|---|---|
| Transformer | 36.0 | 63.6 | 81.6 | 42.3 | 72.7 | 59.3 | - |
| FLASHATTENTION | 37.6 | 63.9 | 81.4 | 43.5 | 72.7 | 59.8 | 2.4× |
| Block-sparse FLASHATTENTION | 37.0 | 63.0 | 81.3 | 43.6 | 73.3 | 59.6 | **2.8×** |
| Linformer [88] | 35.6 | 55.9 | 77.7 | 37.8 | 67.6 | 54.9 | 2.5× |
| Linear Attention [52] | 38.8 | 63.2 | 80.7 | 42.6 | 72.5 | 59.6 | 2.3× |
| Performer [13] | 36.8 | 63.6 | 82.2 | 42.1 | 69.9 | 58.9 | 1.8× |
| Local Attention [83] | 36.1 | 60.2 | 76.7 | 40.6 | 66.6 | 56.0 | 1.7× |
| Reformer [53] | 36.5 | 63.8 | 78.5 | 39.6 | 69.4 | 57.6 | 1.3× |
| Smyrf [20] | 36.1 | 64.1 | 79.0 | 39.6 | 70.5 | 57.9 | 1.7× |

FLASHATTENTION and context length 4K is still 30% faster than GPT-2 from Megatron with context length 1K, while achieving 0.7 better perplexity.

Table 4: GPT-2 small with FLASHATTENTION, with 4× larger context length compared to Megatron-LM, is still 30% faster while achieving 0.7 better perplexity. Training time on 8×A100 GPUs is reported.

| Model implementations | Context length | OpenWebText (ppl) | Training time (speedup) |
|---|---|---|---|
| GPT-2 small - Megatron-LM | 1k | 18.2 | 4.7 days (1.0×) |
| GPT-2 small - FLASHATTENTION | 1k | 18.2 | **2.7 days (1.7×)** |
| GPT-2 small - FLASHATTENTION | 2k | 17.7 | 3.0 days (1.6×) |
| GPT-2 small - FLASHATTENTION | 4k | **17.2** | 3.6 days (1.3×) |

**Long Document Classification.** Training Transformers with longer sequences with FLASHAT-TENTION improves performance on the MIMIC-III [49] and ECtHR [6, 7] datasets. MIMIC-III contains intensive care unit patient discharge summaries, each annotated with multiple labels. ECtHR contains legal cases from the European Court of Human Rights, each of which is mapped to articles of the Convention of Human Rights that were allegedly violaged. Both of these datasets contain very long text documents; the average number of tokens in MIMIC is 2,395 tokens, and the longest document contains 14,562 tokens, while the average and longest numbers in ECtHR are 2,197 and 49,392, respectively. We evaluate lift from increasing the sequence length of a pretrained RoBERTa model [58] (we repeat the positional embeddings, as in Beltagy et al. [3]).

Table 5 shows that sequence length 16K outperforms length 512 by 4.3 points on MIMIC, and that length 8K outperforms length 512 by 8.5 points on ECtHR. The discrepancies may be due to subtle distribution shifts: MIMIC-III contains specialized medical text and thus may be more susceptible to a distribution shift in the document length, whereas ECtHR contains general language.

Table 5: Long Document performance (micro $F_1$) at different sequence lengths using FLASHATTEN-TION.

| | 512 | 1024 | 2048 | 4096 | 8192 | 16384 |
|---|---|---|---|---|---|---|
| MIMIC-III [49] | 52.8 | 50.7 | 51.7 | 54.6 | 56.4 | **57.1** |
| ECtHR [6] | 72.2 | 74.3 | 77.1 | 78.6 | **80.7** | 79.2 |

Table 6: We report the first Transformer model that can achieve non-random performance on Path-X and Path-256.

| Model | Path-X | Path-256 |
|---|---|---|
| Transformer | ✗ | ✗ |
| Linformer [88] | ✗ | ✗ |
| Linear Attention [52] | ✗ | ✗ |
| Performer [13] | ✗ | ✗ |
| Local Attention [83] | ✗ | ✗ |
| Reformer [53] | ✗ | ✗ |
| SMYRF [20] | ✗ | ✗ |
| FLASHATTENTION | **61.4** | ✗ |
| Block-sparse FLASHATTENTION | 56.0 | **63.1** |

**Path-X and Path-256.** The Path-X and Path-256 benchmarks are challenging tasks from the long-range arena benchmark designed to test long context. The task is to classify whether two points in a black and white 128×128 (or 256×256) image have a path connecting them, and the images are fed to the transformer one pixel at a time. In prior work, all transformer models have either run out of memory, or only achieved random performance [83]. There has been a search for alternative architectures that can model such long context [39]. We present here the first result of Transformer models being able to solve Path-X and Path-256 (Table 6). We pretrain a transformer on Path-64, and then transfer to Path-X by spatially interpolating the positional embeddings. FLASHATTENTION achieves 61.4 accuracy on Path-X. Additionally, block-sparse FLASHATTENTION enables the Transformers to scale to sequence length 64K, achieving 63.1 accuracy[4] on Path-256.

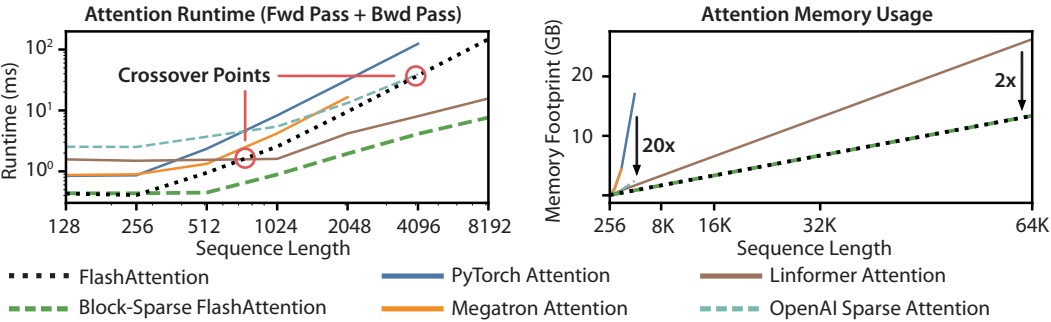

Figure 3: **Left:** runtime of forward pass + backward pass. **Right:** attention memory usage.

### 4.3 Benchmarking Attention

We vary sequence length and measure runtime and memory usage of FLASHATTENTION and block-sparse FLASHATTENTION against various attention baselines on one A100 GPU with 40 GB HBM, with dropout and a padding mask. We compare against reference implementations for exact attention, approximate attention, and sparse attention. We report a subset of baselines in the main body; Appendix E contains more baselines and full details.

**Runtime.** Figure 3 (left) reports the runtime in milliseconds of the forward + backward pass of FLASHATTENTION and block-sparse FLASHATTENTION compared to the baselines in exact, approximate, and sparse attention (exact numbers in Appendix E). Runtime grows quadratically with sequence length, but FLASHATTENTION runs significantly faster than **exact attention** baselines, up to 3× faster than the PyTorch implementation. The runtimes of many approximate/sparse attention mechanisms grow linearly with sequence length, but FLASHATTENTION still runs faster than approximate and sparse attention for short sequences due to fewer memory accesses. The **approximate attention** runtimes begin to cross over with FLASHATTENTION at sequences between 512 and 1024. On the other hand, block-sparse FLASHATTENTION is faster than all implementations of exact, sparse, and approximate attention that we know of, across all sequence lengths.

**Memory Footprint.** Figure 3 (right) shows the memory footprint of FLASHATTENTION and block-sparse FLASHATTENTION compared to various exact, approximate, and sparse attention baselines. FLASHATTENTION and block-sparse FLASHATTENTION have the same memory footprint, which grows linearly with sequence length. FLASHATTENTION is up to 20× more memory efficient than **exact attention** baselines, and is more memory-efficient than the **approximate attention** baselines. All other algorithms except for Linformer run out of memory on an A100 GPU before 64K, and FLASHATTENTION is still 2× more efficient than Linformer.

## 5 Limitations and Future Directions

We discuss limitations of our approach and future directions. Related work is given in Appendix A.

**Compiling to CUDA.** Our current approach to building IO-aware implementations of attention requires writing a new CUDA kernel for each new attention implementation. This requires writing the attention algorithm in a considerably lower-level language than PyTorch, and requires significant engineering effort. Implementations may also not be transferrable across GPU architectures. These limitations suggest the need for a method that supports writing attention algorithms in a high-level language (e.g., PyTorch), and compiling to IO-aware implementations in CUDA—similar to efforts such as Halide in image processing [73].

**IO-Aware Deep Learning.** We believe that the IO-aware approach can extend beyond attention. Attention is the most memory-intensive computation in Transformers, but every layer in a deep network touches GPU HBM. We hope our work inspires IO-aware implementations of additional modules. We discuss these potential extensions in Appendix D.

**Multi-GPU IO-Aware Methods.** Our IO-aware implementation of attention is optimal within constants for computing attention on a single GPU. However, the attention computation may be par-

---

[4]Path-256 requires longer sequences but has relatively shorter paths than Path-X, so it is easier to obtain a higher accuracy.

allelizable across multiple GPUs [75]. Using multiple GPUs adds an additional layer to IO analysis—accounting for data transfer between GPUs. We hope our work inspires future work in this direction.

**Societal Impacts.** As Transformer-based foundation models grow in size and data, our work seeks to understand how to train these large models more efficiently. This may allow a general community with limited access to computational resources to train and understand those foundation models. Our method is applicable to all Transformer-based models, which have a variety of applications, both positive and negative. For example, language modeling may make it easier to spread misinformation, while image classification models may make automatic surveillance easier. Alleviating these risks requires addressing application-specific issues such as privacy, bias, and discrimination.

**Acknowledgments**

Our implementation uses Apex's FMHA code (`https://github.com/NVIDIA/apex/tree/master/apex/contrib/csrc/fmha`) as a starting point. We thank Young-Jun Ko for the in-depth explanation of his FMHA implementation and for his thoughtful answers to our questions about CUDA. We thank Sabri Eyuboglu, Megan Leszczynski, Laurel Orr, Yuhuai Wu, Beidi Chen, and Xun Huang for their constructive feedback and suggestions on early drafts of the paper. We thank Markus Rabe and Charles Staats for helpful discussion of their attention algorithm.

We gratefully acknowledge the support of NIH under No. U54EB020405 (Mobilize), NSF under Nos. CCF1763315 (Beyond Sparsity), CCF1563078 (Volume to Velocity), and 1937301 (RTML); ARL under No. W911NF-21-2-0251 (Interactive Human-AI Teaming); ONR under No. N000141712266 (Unifying Weak Supervision); ONR N00014-20-1-2480: Understanding and Applying Non-Euclidean Geometry in Machine Learning; N000142012275 (NEPTUNE); NXP, Xilinx, LETI-CEA, Intel, IBM, Microsoft, NEC, Toshiba, TSMC, ARM, Hitachi, BASF, Accenture, Ericsson, Qualcomm, Analog Devices, Google Cloud, Salesforce, Total, the HAI-GCP & HAI-Azure Cloud Credits for Research program, the Stanford Data Science Initiative (SDSI), Department of Defense (DoD) through the National Defense Science and Engineering Graduate Fellowship (NDSEG) Program, and members of the Stanford DAWN project: Facebook, Google, and VMWare. The U.S. Government is authorized to reproduce and distribute reprints for Governmental purposes notwithstanding any copyright notation thereon. Any opinions, findings, and conclusions or recommendations expressed in this material are those of the authors and do not necessarily reflect the views, policies, or endorsements, either expressed or implied, of NIH, ONR, or the U.S. Government. Atri Rudra's research is supported by NSF grant CCF-1763481.

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
