# A    Related Work

**IO-Aware Runtime Optimization.** The broad concept of optimizing for reading and writing to fast/slow memory has a long history in computer science and has been known by many names. We draw the most direct connection to the literature of analyzing I/O complexity in this work [1], but concepts of memory hierarchies are fundamental and has appeared in many forms, from the working set model [23], to data locality [90], to the Roofline model of arithmetic intensity [89], to analyses of scalability [61], to standard textbook treatments of computer architecture [42]. We hope that this work encourages the community to adopt these ideas in more parts of the deep learning stack.

**Efficient ML Models with Structured Matrices.** Matrix multiply is the core computational bottleneck of most machine learning models. To reduce the computational complexity, there have been numerous approaches to learn over a more efficient set of matrices. These matrices are called *structured matrices*, which have subquadratic ($o(n^2)$ for dimension $n \times n$) number of parameters and runtime. Most common examples of structured matrices are sparse and low-rank matrices, along with fast transforms commonly encountered in signal processing (Fourier, Chebyshev, sine/cosine, orthogonal polynomials). There have been several more general classes of structured matrices proposed in machine learning: Toeplitz-like [81], low-displacement rank [51], quasi-separable [27]). The butterfly pattern we use for our block-sparse attention is motivated by the fact that butterfly matrices [16, 67] and their products have been shown to be able to express any structured matrices with almost optimal runtime and number of parameters [17, 21]. However, even though structured matrices are efficient in theory, they have not seen wide adoption since it is hard to translate their efficiency to wall-clock speedup since dense unconstrained matrix multiply has very optimize implementation, a phenomenon known as the hardware lottery [43]. Extensions of butterfly matrices [18, 19] aimed to make butterfly matrices more hardware-friendly.

**Sparse Training.** Our block-sparse FLASHATTENTION can be seen as a step towards making sparse model training more efficient. Sparse models have seen success in compressing models for inference (pruning) by sparsifying the weight matrices [25, 40, 41, 57, 79]. For model training, the lottery tickets hypothesis [30–32] suggests that there are a set of small sub-networks derived from a larger dense network that performs as well as the original dense network. Out block-sparse FLASHATTENTION can also be seen as a fixed lottery ticket in the context of attention: we fix the sparsity pattern to be the butterfly pattern through training, and observe that it performs almost as well as the (dense) FLASHATTENTION on the Long-range Arena tasks.

**Efficient Transformer.** Transformer-based models have become the most widely-used architecture in natural language processing [24] and computer vision [26, 95]. However, one of their computational bottlenecks is that their time and memory scales quadratic in the sequence length. There are numerous approaches to overcome this bottleneck, including approximation with hashing (i.e., sparse) such as Reformer [53] and Smyrf [20] and with low-rank approximation such as Performer [13, 56]. One can even combine sparse and low-rank approximation for better accuracy (e.g., Longformer [3], Big-Bird [96], Scatterbrain [9], Long-short transformer [98], Combiner [76]). Other approaches include compressing along the sequence dimension to attend to multiple tokens at once [54, 59, 82, 93]. One can also attend over the states from previous sequences to help lengthen the context (e.g., Transformer-XL [15] and Compressive Transformer [72]). We recommend the survey [84] for more details.

There are several lines of work on developing other modules instead of attention to model longer context. HiPPO [37] and its extensions, most notably S4 [33, 38, 39] projects the history on a polynomial basis, allowing accurate reconstruction of the history through state-space models. They combine the strengths of CNNs (efficient training), RNNs (efficient inference), and continuous models (robust to change in sampling rates). LambdaNetworks [2], AFT [97] and FLASH [44] are other attempts at replacing attention in the context of image classification and language modeling.

# B    Algorithm Details

We first derive the forward and backward passes of attention and show that they can be computed in a memory-efficient manner (requiring extra memory linear instead of quadratic in the sequence length). Though they reduce the amount of extra memory required, naively they still incur quadratic HBM accesses, resulting in slower execution speed. We describe the FLASHATTENTION algorithm to implement both the forward and the backward passes on GPUs that reduces HBM accesses, leading to both faster runtime and smaller memory footprint.

## B.1 Memory-efficient forward pass

The main challenge in making attention memory-efficient is the softmax that couples the columns of $\mathbf{K}$ (and columns of $\mathbf{V}$). Our approach is to compute the softmax normalization constant separately to decouple the columns. This technique [62] has been used in the literature [53, 69] to show that attention computation does not need quadratic *extra* memory (though the number of HBM accesses is still quadratic, resulting in slow run-time).

For simplicity, we omit here the max-shifting step during softmax. The full algorithm in Appendix B.3 contains all the steps.

Recall that given input sequences $\mathbf{Q},\mathbf{K},\mathbf{V} \in \mathbb{R}^{N \times d}$, we want to compute the attention output $\mathbf{O} \in \mathbb{R}^{N \times d}$:
$$\mathbf{S}=\mathbf{Q}\mathbf{K}^\top \in \mathbb{R}^{N \times N}, \quad \mathbf{P}=\text{softmax}(\mathbf{S}) \in \mathbb{R}^{N \times N}, \quad \mathbf{O}=\mathbf{P}\mathbf{V} \in \mathbb{R}^{N \times d}.$$

We have that $S_{ij} = q_i^T k_j$ where $q_i$ and $k_j$ are the $i$-th and $j$-th columns of $\mathbf{Q}$ and $\mathbf{K}$ respectively. Define the normalization constants of softmax:
$$L_i = \sum_j e^{q_i^T k_j}. \tag{1}$$

Let $v_j$ be the $j$-th column of $\mathbf{V}$, then the $i$-th columns of the output is
$$o_i = P_{i:}\mathbf{V} = \sum_j P_{ij} v_j = \sum_j \frac{e^{q_i^T k_j}}{L_i} v_j. \tag{2}$$

We see that once $L_i$ is computed, we can compute $o_i$ without extra memory by repeatedly summing $\frac{e^{q_i^T k_j}}{L_i} v_j$. Therefore the forward pass can be computed with $O(n)$ extra memory:

1. Compute $L_i$ for all $i$ according to Eq. (1), which takes $O(n)$ extra memory.
2. Compute $o_i$ for all $i$ according to Eq. (2), which takes $O(d)$ extra memory.

## B.2 Memory-efficient backward pass

We derive the backward pass of attention and show that it can also be computed with linear memory. Rabe and Staats [69] suggests that the backward pass can be done without quadratic extra memory by applying gradient checkpointing to the memory-efficient forward pass. We instead derive the backward pass explicitly and show how it can be computed in a memory-efficient manner.

Suppose that there is a scalar loss function $\phi$, and let the output gradient be $\mathbf{dO} \in \mathbb{R}^{n \times d}$ (where $\mathbf{dO}$ denotes $\frac{\partial \phi}{\partial \mathbf{O}}$). We want to compute the input gradients $\mathbf{dQ},\mathbf{dK},\mathbf{dV} \in \mathbb{R}^{n \times d}$ (where $\mathbf{dQ},\mathbf{dK},\mathbf{dV}$ denote $\frac{\partial \phi}{\partial \mathbf{Q}}, \frac{\partial \phi}{\partial \mathbf{K}}, \frac{\partial \phi}{\partial \mathbf{V}}$ respectively).

The gradient $\mathbf{dV}$ is easy to see. Applying reverse-mode autodiff by hand (aka the chain rule), we obtain (in matrix notation) $\mathbf{dV} = \mathbf{P}^T \mathbf{dO}$. Thus:
$$dv_j = \sum_i P_{ij} do_i = \sum_i \frac{e^{q_i^T k_j}}{L_i} do_i. \tag{3}$$

Since we already computed $L_i$, $dv_j$ can be computed without extra memory by repeated summing.

The gradients $\mathbf{dQ}$ and $\mathbf{dK}$ are a little more complicated. We go through the gradients $\mathbf{dP}$ and $\mathbf{dS}$ first. From Eq. (2), we have that $\mathbf{dP} = \mathbf{dO}\mathbf{V}^T$, and so:
$$dP_{ij} = do_i^T v_j.$$

Recall that $P_{i:} = \text{softmax}(S_{i:})$. Using the fact that the Jacobian of $y = \text{softmax}(x)$ is $\text{diag}(y) - yy^T$, we have that
$$dS_{i:} = (\text{diag}(P_{i:}) - P_{i:}P_{i:}^T) dP_{i:} = P_{i:} \circ dP_{i:} - (P_{i:}^T dP_{i:}) P_{i:},$$
where $\circ$ denotes pointwise multiplication.

Define
$$D_i = P_{i:}^T dP_{i:} = \sum_j \frac{e^{q_i^T k_j}}{L_i} do_i^T v_j = do_i^T \sum_j \frac{e^{q_i^T k_j}}{L_i} v_j = do_i^T o_i, \tag{4}$$

then
$$dS_{i:} = P_{i:} \circ dP_{i:} - D_i P_{i:}.$$

Hence
$$dS_{ij} = P_{ij} dP_{ij} - D_i P_{ij} = P_{ij}(dP_{ij} - D_i).$$

Now we can get the gradients **dQ** and **dK**. Recall that $S_{ij} = q_i^T k_j$, so

$$dq_i = \sum_j dS_{ij} k_j = \sum_j P_{ij}(dP_{ij} - D_i)k_j = \sum_j \frac{e^{q_i^T k_j}}{L_i}(do_i^T v_j - D_i)k_j. \qquad (5)$$

Similarly,

$$dk_j = \sum_i dS_{ij} q_i = \sum_i P_{ij}(dP_{ij} - D_i)q_i = \sum_i \frac{e^{q_i^T k_j}}{L_i}(do_i^T v_j - D_i)q_i. \qquad (6)$$

Therefore the backward pass can also be computed with $O(n)$ extra memory:

1. Compute $dv_j$ for all $j$ according to Eq. (3), which takes $O(d)$ extra memory.
2. Compute $D_i$ for all $i$ according to Eq. (4), which takes $O(n)$ extra memory.
3. Compute $dq_i$ for all $i$ according to Eq. (5), which takes $O(d)$ extra memory.
4. Compute $dk_j$ for all $j$ according to Eq. (6), which takes $O(d)$ extra memory.

## B.3  FLASHATTENTION: Forward Pass

We describe the full details of FLASHATTENTION forward pass. Given input sequences $\mathbf{Q}, \mathbf{K}, \mathbf{V} \in \mathbb{R}^{N \times d}$, we want to compute the attention output $\mathbf{O} \in \mathbb{R}^{N \times d}$:

$$\mathbf{S} = \tau \mathbf{Q} \mathbf{K}^\top \in \mathbb{R}^{N \times N}, \quad \mathbf{S}^{\text{masked}} = \text{MASK}(S) \in \mathbb{R}^{N \times N}, \quad \mathbf{P} = \text{softmax}(\mathbf{S}^{\text{masked}}) \in \mathbb{R}^{N \times N},$$

$$\mathbf{P}^{\text{dropped}} = \text{dropout}(\mathbf{P}, p_{\text{drop}}), \quad \mathbf{O} = \mathbf{P}^{\text{dropped}} \mathbf{V} \in \mathbb{R}^{N \times d},$$

where $\tau \in \mathbb{R}$ is some softmax scaling (typically $\frac{1}{\sqrt{d}}$), MASK is some masking function that sets some entries of the input to $-\infty$ and keep other entries the same (e.g., key padding mask when sequences in the batch don't have the same lengths and are padded), and dropout$(x, p)$ applies dropout to $x$ elementwise (i.e., output $\frac{x}{1-p}$ with probability $1-p$ and output $0$ with probability $p$ for each element $x$).

The full algorithm is in Algorithm 2. We save the output $\mathbf{O}$, the softmax statistics $\ell$ and $m$, and the pseudo-random number generator state $\mathcal{R}$ for the backward pass.

---

**Algorithm 2** FLASHATTENTION Forward Pass

---

**Require:** Matrices $\mathbf{Q}, \mathbf{K}, \mathbf{V} \in \mathbb{R}^{N \times d}$ in HBM, on-chip SRAM of size $M$, softmax scaling constant $\tau \in \mathbb{R}$, masking function MASK, dropout probability $p_{\text{drop}}$.

1: Initialize the pseudo-random number generator state $\mathcal{R}$ and save to HBM.
2: Set block sizes $B_c = \left\lceil \frac{M}{4d} \right\rceil, B_r = \min\left(\left\lceil \frac{M}{4d} \right\rceil, d\right)$.
3: Initialize $\mathbf{O} = (0)_{N \times d} \in \mathbb{R}^{N \times d}, \ell = (0)_N \in \mathbb{R}^N, m = (-\infty)_N \in \mathbb{R}^N$ in HBM.
4: Divide $\mathbf{Q}$ into $T_r = \left\lceil \frac{N}{B_r} \right\rceil$ blocks $\mathbf{Q}_1, ..., \mathbf{Q}_{T_r}$ of size $B_r \times d$ each, and divide $\mathbf{K}, \mathbf{V}$ in to $T_c = \left\lceil \frac{N}{B_c} \right\rceil$ blocks $\mathbf{K}_1, ..., \mathbf{K}_{T_c}$ and $\mathbf{V}_1, ..., \mathbf{V}_{T_c}$, of size $B_c \times d$ each.
5: Divide $\mathbf{O}$ into $T_r$ blocks $\mathbf{O}_i, ..., \mathbf{O}_{T_r}$ of size $B_r \times d$ each, divide $\ell$ into $T_r$ blocks $\ell_i, ..., \ell_{T_r}$ of size $B_r$ each, divide $m$ into $T_r$ blocks $m_1, ..., m_{T_r}$ of size $B_r$ each.
6: **for** $1 \le j \le T_c$ **do**
7:     Load $\mathbf{K}_j, \mathbf{V}_j$ from HBM to on-chip SRAM.
8:     **for** $1 \le i \le T_r$ **do**
9:         Load $\mathbf{Q}_i, \mathbf{O}_i, \ell_i, m_i$ from HBM to on-chip SRAM.
10:        On chip, compute $\mathbf{S}_{ij} = \tau \mathbf{Q}_i \mathbf{K}_j^T \in \mathbb{R}^{B_r \times B_c}$.
11:        On chip, compute $\mathbf{S}_{ij}^{\text{masked}} = \text{MASK}(\mathbf{S}_{ij})$.
12:        On chip, compute $\tilde{m}_{ij} = \text{rowmax}(\mathbf{S}_{ij}^{\text{masked}}) \in \mathbb{R}^{B_r}, \tilde{\mathbf{P}}_{ij} = \exp(\mathbf{S}_{ij}^{\text{masked}} - \tilde{m}_{ij}) \in \mathbb{R}^{B_r \times B_c}$ (pointwise), $\tilde{\ell}_{ij} = \text{rowsum}(\tilde{\mathbf{P}}_{ij}) \in \mathbb{R}^{B_r}$.
13:        On chip, compute $m_i^{\text{new}} = \max(m_i, \tilde{m}_{ij}) \in \mathbb{R}^{B_r}, \ell_i^{\text{new}} = e^{m_i - m_i^{\text{new}}} \ell_i + e^{\tilde{m}_{ij} - m_i^{\text{new}}} \tilde{\ell}_{ij} \in \mathbb{R}^{B_r}$.
14:        On chip, compute $\tilde{\mathbf{P}}_{ij}^{\text{dropped}} = \text{dropout}(\tilde{\mathbf{P}}_{ij}, p_{\text{drop}})$.
15:        Write $\mathbf{O}_i \leftarrow \text{diag}(\ell_i^{\text{new}})^{-1}(\text{diag}(\ell_i) e^{m_i - m_i^{\text{new}}} \mathbf{O}_i + e^{\tilde{m}_{ij} - m_i^{\text{new}}} \tilde{\mathbf{P}}_{ij}^{\text{dropped}} \mathbf{V}_j)$ to HBM.
16:        Write $\ell_i \leftarrow \ell_i^{\text{new}}, m_i \leftarrow m_i^{\text{new}}$ to HBM.
17:    **end for**
18: **end for**
19: Return $\mathbf{O}, \ell, m, \mathcal{R}$.

---

## B.4 FLASHATTENTION: Backward Pass

We describe the full details of FLASHATTENTION backward pass. Given input sequences $\mathbf{Q},\mathbf{K},\mathbf{V} \in \mathbb{R}^{N \times d}$, the output $\mathbf{O} \in \mathbb{R}^{N \times d}$, and the output gradient $\mathbf{dO}$, we want to compute the input gradients $\mathbf{dQ},\mathbf{dK},\mathbf{dV} \in \mathbb{R}^{N \times d}$.

We first describe the standard attention backward pass in Algorithm 3 for completeness.

---

**Algorithm 3** Standard Attention Backward Pass

---

**Require:** Matrices $\mathbf{Q},\mathbf{K},\mathbf{V},\mathbf{dO} \in \mathbb{R}^{N \times d}$, $\mathbf{P} \in \mathbb{R}^{N \times N}$ in HBM.
 1: Load $\mathbf{P},\mathbf{dO}$ by blocks from HBM, compute $\mathbf{dV}=\mathbf{P}^\top \mathbf{dO} \in \mathbb{R}^{N \times d}$, write $\mathbf{dV}$ to HBM.
 2: Load $\mathbf{dO},\mathbf{V}$ by blocks from HBM, compute $\mathbf{dP}=\mathbf{dO}\mathbf{V}^\top \in \mathbb{R}^{N \times N}$, write $\mathbf{dP}$ to HBM.
 3: Read $\mathbf{P},\mathbf{dP}$ from HBM, compute $\mathbf{dS} \in \mathbb{R}^{N \times N}$ where $dS_{ij} = P_{ij}(dP_{ij} - \sum_l P_{il}dP_{il})$, write $\mathbf{dS}$ to HBM.
 4: Load $\mathbf{dS}$ and $\mathbf{K}$ by blocks from HBM, compute $\mathbf{dQ}=\mathbf{dS}\mathbf{K}$, write $\mathbf{dQ}$ to HBM.
 5: Load $\mathbf{dS}$ and $\mathbf{Q}$ by blocks from HBM, compute $\mathbf{dK}=\mathbf{dS}^\top \mathbf{Q}$, write $\mathbf{dK}$ to HBM.
 6: Return $\mathbf{dQ},\mathbf{dK},\mathbf{dV}$.

---

We now make two observations about FLASHATTENTION backward pass:

1. We do not need to store the dropout mask of size $O(N^2)$ from the forward pass. Instead, we can save the pseudo-random number generator states from the forward pass and re-generate the dropout mask in the backward pass. This allows us to only use $O(N)$ extra memory.

2. When computing the softmax gradient, we use Eq. (4) to compute $D_i = P_{i:}^\top dP_{i:}$ without reducing over $P_{i:}$ and $dP_{i:}$ of size $N$ (they might not fit into SRAM). Instead we can rewrite $D_i = do_i^\top o_i$ and compute the dot product between vectors of size $d$.

The full FLASHATTENTION backward pass algorithm is in Algorithm 4. Conceptually it is just a block version of the derivation in Appendix B.2.

We see that similar to the forward pass, the backward pass performs $O(N^2)$ FLOPs and only requires $O(N)$ extra memory beyond inputs, output, output gradient, and input gradients.

We analyze the IO-complexity of the backward pass, similar to the forward pass (Theorem 2).

**Theorem 5.** *Let $N$ be the sequence length, $d$ be the head dimension, and $M$ be size of SRAM with $d \leq M \leq Nd$. Standard attention (Algorithm 0) backward pass requires $\Theta(Nd+N^2)$ HBM accesses, while* FLASHATTENTION *backward pass (Algorithm 4) requires $\Theta(N^2d^2M^{-1})$ HBM accesses.*

The proof is in Appendix C.

## B.5 Comparison with Rabe and Staats [69]

We describe here some similarities and differences between our FLASHATTENTION algorithm and the algorithm of Rabe and Staats [69].

Conceptually, both FLASHATTENTION and Rabe and Staats [69] operate on blocks of the attention matrix using the well-established technique of tiling (or softmax scaling) [53, 62]. To reduce the memory footprint, both methods avoid storing the large attention matrix in the forward pass and recompute it in the backward pass.

The first major difference is that Rabe and Staats [69] focuses on the reducing the total memory footprint (maximum amount of GPU memory required) while FLASHATTENTION focuses on reducing memory accesses (the number of memory reads/writes). As mentioned in Section 2, the amount of memory access is the primary determining factor of runtime. Reducing memory accesses also necessarily reduces the total amount of memory required (e.g., if an operation incurs $A$ memory accesses, then its total memory requirement is at most $A$). As a result, FLASHATTENTION is faster than standard attention (2-4×) while Rabe and Staats [69] is around the same speed or slightly slower than standard attention. In terms of total memory required, both methods offer substantial memory saving.

The second difference between the two methods is the way information is summarized from each block to pass to the next block. Rabe and Staats [69] summarizes each block with its temporary output along with the softmax normalization statistics. At the end of the forward pass, the temporary outputs of all the blocks are combined using the statistics to produce the final output. FLASHATTENTION

---

**Algorithm 4** FLASHATTENTION Backward Pass

---

**Require:** Matrices $\mathbf{Q},\mathbf{K},\mathbf{V},\mathbf{O},\mathbf{dO} \in \mathbb{R}^{N \times d}$ in HBM, vectors $\ell, m \in \mathbb{R}^N$ in HBM, on-chip SRAM of size $M$, softmax scaling constant $\tau \in \mathbb{R}$, masking function MASK, dropout probability $p_{\text{drop}}$, pseudo-random number generator state $\mathcal{R}$ from the forward pass.

1:  Set the pseudo-random number generator state to $\mathcal{R}$.
2:  Set block sizes $B_c = \left\lceil \frac{M}{4d} \right\rceil, B_r = \min\left( \left\lceil \frac{M}{4d} \right\rceil, d \right)$.
3:  Divide $\mathbf{Q}$ into $T_r = \left\lceil \frac{N}{B_r} \right\rceil$ blocks $\mathbf{Q}_1, ..., \mathbf{Q}_{T_r}$ of size $B_r \times d$ each, and divide $\mathbf{K}, \mathbf{V}$ in to $T_c = \left\lceil \frac{N}{B_c} \right\rceil$ blocks $\mathbf{K}_1, ..., \mathbf{K}_{T_c}$ and $\mathbf{V}_1, ..., \mathbf{V}_{T_c}$, of size $B_c \times d$ each.
4:  Divide $\mathbf{O}$ into $T_r$ blocks $\mathbf{O}_i, ..., \mathbf{O}_{T_r}$ of size $B_r \times d$ each, divide $\mathbf{dO}$ into $T_r$ blocks $\mathbf{dO}_i, ..., \mathbf{dO}_{T_r}$ of size $B_r \times d$ each, divide $\ell$ into $T_r$ blocks $\ell_i, ..., \ell_{T_r}$ of size $B_r$ each, divide $m$ into $T_r$ blocks $m_1, ..., m_{T_r}$ of size $B_r$ each.
5:  Initialize $\mathbf{dQ} = (0)_{N \times d}$ in HBM and divide it into $T_r$ blocks $\mathbf{dQ}_1, ..., \mathbf{dQ}_{T_r}$ of size $B_r \times d$ each. Initialize $\mathbf{dK} = (0)_{N \times d}, \mathbf{dV} = (0)_{N \times d}$ in HBM and divide $\mathbf{dK}, \mathbf{dV}$ in to $T_c$ blocks $\mathbf{dK}_1, ..., \mathbf{dK}_{T_c}$ and $\mathbf{dV}_1, ..., \mathbf{dV}_{T_c}$, of size $B_c \times d$ each.
6:  **for** $1 \le j \le T_c$ **do**
7:      Load $\mathbf{K}_j, \mathbf{V}_j$ from HBM to on-chip SRAM.
8:      Initialize $\mathbf{d\tilde{K}}_j = (0)_{B_c \times d}, \mathbf{d\tilde{V}}_j = (0)_{B_c \times d}$ on SRAM.
9:      **for** $1 \le i \le T_r$ **do**
10:         Load $\mathbf{Q}_i, \mathbf{O}_i, \mathbf{dO}_i, \mathbf{dQ}_i, \ell_i, m_i$ from HBM to on-chip SRAM.
11:         On chip, compute $\mathbf{S}_{ij} = \tau \mathbf{Q}_i \mathbf{K}_j^T \in \mathbb{R}^{B_r \times B_c}$.
12:         On chip, compute $\mathbf{S}_{ij}^{\text{masked}} = \text{MASK}(\mathbf{S}_{ij})$.
13:         On chip, compute $\mathbf{P}_{ij} = \text{diag}(l_i)^{-1} \exp(\mathbf{S}_{ij}^{\text{masked}} - m_i) \in \mathbb{R}^{B_r \times B_c}$.
14:         On chip, compute dropout mask $\mathbf{Z}_{ij} \in \mathbb{R}^{B_r \times B_c}$ where each entry has value $\frac{1}{1 - p_{\text{drop}}}$ with probability $1 - p_{\text{drop}}$ and value $0$ with probability $p_{\text{drop}}$.
15:         On chip, compute $\mathbf{P}_{ij}^{\text{dropped}} = \mathbf{P}_{ij} \circ \mathbf{Z}_{ij}$ (pointwise multiply).
16:         On chip, compute $\mathbf{d\tilde{V}}_j \leftarrow \mathbf{d\tilde{V}}_j + (\mathbf{P}_{ij}^{\text{dropped}})^\top \mathbf{dO}_i \in \mathbb{R}^{B_c \times d}$.
17:         On chip, compute $\mathbf{dP}_{ij}^{\text{dropped}} = \mathbf{dO}_i \mathbf{V}_j^\top \in \mathbb{R}^{B_r \times B_c}$.
18:         On chip, compute $\mathbf{dP}_{ij} = \mathbf{dP}_{ij}^{\text{dropped}} \circ \mathbf{Z}_{ij}$ (pointwise multiply).
19:         On chip, compute $D_i = \text{rowsum}(\mathbf{dO}_i \circ \mathbf{O}_i) \in \mathbb{R}^{B_r}$.
20:         On chip, compute $\mathbf{dS}_{ij} = \mathbf{P}_{ij} \circ (\mathbf{dP}_{ij} - D_i) \in \mathbb{R}^{B_r \times B_c}$.
21:         Write $\mathbf{dQ}_i \leftarrow \mathbf{dQ}_i + \tau \mathbf{dS}_{ij} \mathbf{K}_j \in \mathbb{R}^{B_r \times d}$ to HBM.
22:         On chip, compute $\mathbf{d\tilde{K}}_j \leftarrow \mathbf{d\tilde{K}}_j + \tau \mathbf{dS}_{ij}^\top \mathbf{Q}_i \in \mathbb{R}^{B_c \times d}$.
23:     **end for**
24:     Write $\mathbf{dK}_j \leftarrow \mathbf{d\tilde{K}}_j, \mathbf{dV}_j \leftarrow \mathbf{d\tilde{V}}_j$ to HBM.
25: **end for**
26: Return $\mathbf{dQ}, \mathbf{dK}, \mathbf{dV}$.

---

instead incrementally updates the output (Algorithm 1 line 1) after processing each block, so only one copy of the output is needed (instead of $K$ copies for $K$ blocks). This means that FLASHATTENTION has smaller total memory requirement compared to Rabe and Staats [69].

The final major difference is the way the backward pass is computed. Rabe and Staats [69] uses gradient checkpointing to recompute the attention matrix and the temporary output of each block. FLASHATTENTION instead simplifies the backward pass analytically (Appendices B.2 and B.4). It only recomputes the attention matrix and does not recompute the temporary output of each block. This reduces the memory requirement for the backward pass and yields speedup.

## C  Proofs

*Proof of Theorem 1.* We first count the number of FLOPs and extra memory required.

The dominating FLOPs are from matrix multiplication. In the inner loop, (Algorithm 1 line 1), we compute $\mathbf{Q}_i \mathbf{K}_j^\top \in \mathbb{R}^{B_r \times B_c}$ for $\mathbf{Q}_i \in \mathbb{R}^{B_r \times d}$ and $\mathbf{K}_j \in \mathbb{R}^{B_c \times d}$, which takes $O(B_r B_c d)$ FLOPs. We also compute (Algorithm 1 line 1) $\mathbf{\tilde{P}}_{ij} \mathbf{V}_j \in \mathbb{R}^{B_r \times d}$ for $\mathbf{\tilde{P}}_{ij} \in \mathbb{R}^{B_r \times B_c}$ and $\mathbf{V}_j \in \mathbb{R}^{B_c \times d}$, which takes $O(B_r B_c d)$

FLOPs. We execute the inner loops $T_c T_r = \left\lceil \frac{N}{B_c} \right\rceil \left\lceil \frac{N}{B_r} \right\rceil$ times. Therefore the total number of FLOPs is

$$O\left(\frac{N^2}{B_c B_r} B_r B_c d\right) = O(N^2 d).$$

In terms of extra memory required, we see that we need $O(N)$ memory to store the statistics $(\ell, m)$.

We now prove the algorithm's correctness by induction on $j$ for $0 \le j \le T_c$. Let $\mathbf{K}_{:j} \in \mathbb{R}^{jB_c \times d}$ be the first $jB_c$ rows of $\mathbf{K}$, and similarly $\mathbf{V}_{:j} \in \mathbb{R}^{jB_c \times d}$ the the first $jB_c$ rows of $\mathbf{V}$. Let $\mathbf{S}_{:,:j} = \mathbf{Q}\mathbf{K}_{:j}^\top \in \mathbb{R}^{N \times jB_c}$, and $\mathbf{P}_{:,:j} = \text{softmax}(\mathbf{S}_{:,:j}) \in \mathbb{R}^{N \times jB_c}$ (softmax applied row-wise). Let $m^j, \ell^{(j)}, \mathbf{O}^{(j)}$ be the values of $m, \ell, \mathbf{O}$ in HBM after the $j$-th iteration of the outer loop (Algorithm 1 line 1). (Note that these values of $m, \ell, \mathbf{O}$ are updated after each iteration of the outer loop.) We want to show that after the $j$-th iteration of the outer loop, we have computed in HBM:

$$m^{(j)} = \text{rowmax}(\mathbf{S}_{:,:j}) \in \mathbb{R}^N, \quad \ell^{(j)} = \text{rowsum}(\exp(\mathbf{S}_{:,:j} - m^{(j)})) \in \mathbb{R}^N, \quad \mathbf{O}^{(j)} = \mathbf{P}_{:,:j} \mathbf{V}_{:j} \in \mathbb{R}^{N \times d}.$$

Based on our initialization (Algorithm 1 line 1), this claim is true for $j = 0$ (i.e., before the any iteration of the outer loop is executed). Suppose that the claim holds for some $j = 0, \ldots, T_c - 1$. We want to show that the claim also holds for $j + 1$. Indeed, when we update the statistics in the inner loop (Algorithm 1 line 1) on the $(j+1)$-th iteration of the outer loop, we update $m^{(j+1)} = \max(m^{(j)}, \tilde{m})$ where $\tilde{m} \in \mathbb{R}^N$ is the row-max of $\mathbf{S}_{:,j:j+1}$, the slice of $\mathbf{S}$ from column $jB_c$ to column $(j+1)B_c - 1$. This implies that

$$m^{(j+1)} = \text{rowmax}(\mathbf{S}_{:,:j+1}) \in \mathbb{R}^N.$$

Similarly, we update

$$\ell^{(j+1)} = e^{m^{(j)} - m^{(j+1)}} \ell^{(j)} + e^{\tilde{m} - m^{(j+1)}} \tilde{\ell},$$

where $\tilde{\ell} = \text{rowsum}(\exp(\mathbf{S}_{:,j:j+1} - \tilde{m})) \in \mathbb{R}^N$. By the same algebraic manipulation in Section 3.1, we obtain:

$$\ell^{(j+1)} = \text{rowsum}(\exp(\mathbf{S}_{:,:j+1} - m^{(j+1)})) \in \mathbb{R}^N.$$

Let $\mathbf{V}_{j:j+1}$ be the slice of $\mathbf{V}$ from column $jB_c$ to column $(j+1)B_c - 1$, we also update:

$$\mathbf{O}^{(j+1)} = \text{diag}(\ell^{(j+1)})^{-1}(\text{diag}(\ell^{(j)}) e^{m^{(j)} - m^{(j+1)}} \mathbf{O}^{(j)} + e^{\tilde{m} - m^{(j+1)}} \exp(\mathbf{S}_{j:j+1} - \tilde{m}) \mathbf{V}_{j:j+1})$$

$$= \text{diag}(\ell^{(j+1)})^{-1}(\text{diag}(\ell^{(j)}) e^{m^{(j)} - m^{(j+1)}} \mathbf{P}_{:,:j} \mathbf{V}_{:j} + e^{-m^{(j+1)}} \exp(\mathbf{S}_{j:j+1}) \mathbf{V}_{j:j+1})$$

$$= \text{diag}(\ell^{(j+1)})^{-1}(\text{diag}(\ell^{(j)}) e^{m^{(j)} - m^{(j+1)}} \text{diag}(\ell^{(j)}) \exp(\mathbf{S}_{:,:j} - m^{(j)}) \mathbf{V}_{:j} + e^{-m^{(j+1)}} \exp(\mathbf{S}_{j:j+1}) \mathbf{V}_{j:j+1})$$

$$= \text{diag}(\ell^{(j+1)})^{-1}(e^{-m^{(j+1)}} \exp(\mathbf{S}_{:,:j}) \mathbf{V}_{:j} + e^{-m^{(j+1)}} \exp(\mathbf{S}_{j:j+1}) \mathbf{V}_{j:j+1})$$

$$= \text{diag}(\ell^{(j+1)})^{-1}(\exp(\mathbf{S}_{:,:j} - m^{(j+1)}) \mathbf{V}_{:j} + \exp(\mathbf{S}_{j:j+1} - m^{(j+1)}) \mathbf{V}_{j:j+1})$$

$$= \text{diag}(\ell^{(j+1)})^{-1}\left(\exp\left(\begin{bmatrix} \mathbf{S}_{:,:j} & \mathbf{S}_{j:j+1} \end{bmatrix} - m^{(j+1)}\right)\right)\begin{bmatrix} \mathbf{V}_{:j} \\ \mathbf{V}_{j:j+1} \end{bmatrix}$$

$$= \text{softmax}(\mathbf{S}_{:j+1}) \mathbf{V}_{:j+1}.$$

We then see that the claim is also true for $j + 1$. By induction, the claim is true for all $j = 0, \ldots, T_c$.

When $j = T_c$, we conclude that the final value of $\mathbf{O}$ in HBM is $\text{softmax}(\mathbf{S})\mathbf{V} = \text{softmax}(\mathbf{Q}\mathbf{K}^\top)\mathbf{V}$.

$\square$

*Proof of Theorem 2.* We first analyze the IO complexity of standard attention implementation. The inputs $\mathbf{Q}, \mathbf{K}, \mathbf{V} \in \mathbb{R}^{N \times d}$ reside in HBM, and the at the end of the algorithm the output $\mathbf{O} \in \mathbb{R}^{N \times d}$ is written to HBM.

In the first step of computing the matrix multiply $\mathbf{S} = \mathbf{Q}\mathbf{K}^\top$, the inputs $\mathbf{Q}, \mathbf{K}$ are read from HBM and the output $\mathbf{S} \in \mathbb{R}^{N \times N}$ is written to HBM (Algorithm 0 line 0). This incurs $\Theta(Nd + N^2)$ HBM accesses.

In the second step of computing $\mathbf{P} = \text{softmax}(\mathbf{S})$, the input $\mathbf{S}$ is read from HBM and the output $\mathbf{P}$ is written to HBM (Algorithm 0 line 0). This incurs $\Theta(N^2)$ HBM accesses.

In the last step of computing $\mathbf{O} = \mathbf{P}\mathbf{V}$, the inputs $\mathbf{P}, \mathbf{V}$ are read from global memory and the output $\mathbf{O}$ is written to HBM (Algorithm 0 line 0). This incurs $\Theta(Nd + N^2)$ HBM accesses.

Overall, standard attention implementation requires $\Theta(Nd + N^2)$ global memory accesses.

We now analyze the IO complexity of streaming attention.

Following Algorithm 1, we see that each element of $\mathbf{K}$ and $\mathbf{V}$ is loaded from HBM once (Algorithm 1 line 1). We make $T_c$ passes over $\mathbf{Q}$ and $\mathbf{O}$, each pass loading all of $\mathbf{Q}$ and all of $\mathbf{O}$ to HBM (Algorithm 1 line 1). Therefore the number of HBM accesses is $\Theta(Nd + NdT_c) = \Theta(NdT_c)$.

We derive the conditions on the block sizes $B_c$ and $B_r$. We need the blocks $\mathbf{K}_j$ and $\mathbf{V}_j$ of size $B_c \times d$ to fit into on-chip memory, which translates to:

$$B_c d = O(M) \Leftrightarrow B_c = O\left(\frac{M}{d}\right).$$

Similarly, we need the blocks $\mathbf{Q}_i, \mathbf{O}_i$ of size $B_r \times d$ to fit into on-chip memory, which translates to:

$$B_r d = O(M) \Leftrightarrow B_r = O\left(\frac{M}{d}\right).$$

Finally, we need the block $\mathbf{S}_{ij}$ of size $B_r \times B_c$ to fit into on-chip memory, which translates to:

$$B_r B_c = O(M).$$

We therefore set:

$$B_c = \Theta\left(\frac{M}{d}\right), \qquad B_r = \Theta\left(\min\left(\frac{M}{d}, \frac{M}{B_c}\right)\right) = \Theta\left(\min\left(\frac{M}{d}, d\right)\right).$$

We then have:

$$T_c = \frac{N}{B_c} = \Theta\left(\frac{Nd}{M}\right).$$

As a result, the number of HBM accesses is:

$$\Theta(NdT_c) = \Theta\left(\frac{N^2 d^2}{M}\right).$$

□

*Proof of Proposition 3.* For contradiction, suppose that there exists an algorithm that computes exact attention where the number for HBM access for all $M \in [d, Nd]$ is

$$o\left(\frac{N^2 d^2}{M}\right).$$

In the regime of $M = \Theta(Nd)$, this results in the number of HBM accesses:

$$o\left(\frac{N^2 d^2}{Nd}\right) = o(Nd).$$

However, the input to attention (matrices $\mathbf{Q}, \mathbf{K}, \mathbf{V}$) and the output $\mathbf{O}$ have size $Nd$ and they start out being in HBM, so if the algorithm computes exact attention it must incur at least $\Omega(Nd)$ HBM accesses. This is a contradiction. □

*Proof of Theorem 5.* The IO complexity of the attention backward is very similar to the IO complexity of the attention forward (Theorem 2). Here we provide a sketch of the proof.

We first analyze the IO complexity of standard attention backward pass. The inputs $\mathbf{Q}, \mathbf{K}, \mathbf{V}, \mathbf{dO} \in \mathbb{R}^{N \times d}$ reside in HBM, and the at the end of the algorithm the outputs $\mathbf{dQ}, \mathbf{dK}, \mathbf{dV} \in \mathbb{R}^{N \times d}$ are written to HBM.

At each step of the standard attention backward pass, one needs to load inputs of size $Nd$ or $N^2$ from HBM, and needs to write the outputs of size $N^2$ or $Nd$ to HBM. This incurs $\Theta(Nd + N^2)$ HBM accesses.

We now analyze the IO complexity of FLASHATTENTION backward pass.

Similar to Theorem 2, we see that each element of $\mathbf{K}$ and $\mathbf{V}$ is loaded from HBM once. Each element of $\mathbf{dK}$ and $\mathbf{dV}$ is only written to HBM once. We make $T_c$ passes over $\mathbf{Q}, \mathbf{O}, \mathbf{dO}$, each pass loading all of $\mathbf{Q}, \mathbf{O}, \mathbf{dO}$ to HBM. We also make $T_c$ passes over $\mathbf{dQ}$, each pass reading/writing all of $\mathbf{dQ}$ from/to HBM. Therefore the number of HBM accesses is $\Theta(Nd + NdT_c) = \Theta(NdT_c)$.

As in the proof of Theorem 2, the constraints on the block sizes are that:

$$B_c = \Theta\left(\frac{M}{d}\right), \qquad B_r = \Theta\left(\min\left(\frac{M}{d}, d\right)\right).$$

We then have:

$$T_c = \frac{N}{B_c} = \Theta\left(\frac{Nd}{M}\right).$$

As a result, the number of HBM accesses is:

$$\Theta(NdT_c) = \Theta\left(\frac{N^2 d^2}{M}\right).$$

□

# D  Extension Details

## D.1  Block-sparse FLASHATTENTION

We describe the full block-sparse FLASHATTENTION algorithm in Algorithm 5. The algorithm is identical to Algorithm 2, except that we skip zero blocks.

---

**Algorithm 5** Block-Sparse FLASHATTENTION Forward Pass

---

**Require:** Matrices $\mathbf{Q},\mathbf{K},\mathbf{V} \in \mathbb{R}^{N \times d}$ in HBM, on-chip SRAM of size $M$, softmax scaling constant $\tau \in \mathbb{R}$, masking function MASK, dropout probability $p_{\text{drop}}$, block sizes $B_c = \left\lceil \frac{M}{4d} \right\rceil, B_r = \min\left(\left\lceil \frac{M}{4d} \right\rceil, d\right)$, block sparsity mask $M \in \{0,1\}^{N/B_r \times N/B_c}$..
1: Initialize the pseudo-random number generator state $\mathcal{R}$ and save to HBM.
2: Initialize $\mathbf{O} = (0)_{N \times d} \in \mathbb{R}^{N \times d}, \ell = (0)_N \in \mathbb{R}^N, m = (-\infty)_N \in \mathbb{R}^N$ in HBM.
3: Divide $\mathbf{Q}$ into $T_r = \left\lceil \frac{N}{B_r} \right\rceil$ blocks $\mathbf{Q}_1, ..., \mathbf{Q}_{T_r}$ of size $B_r \times d$ each, and divide $\mathbf{K}, \mathbf{V}$ in to $T_c = \left\lceil \frac{N}{B_c} \right\rceil$ blocks $\mathbf{K}_1, ..., \mathbf{K}_{T_c}$ and $\mathbf{V}_1, ..., \mathbf{V}_{T_c}$, of size $B_c \times d$ each.
4: Divide $\mathbf{O}$ into $T_r$ blocks $\mathbf{O}_i, ..., \mathbf{O}_{T_r}$ of size $B_r \times d$ each, divide $\ell$ into $T_r$ blocks $\ell_i, ..., \ell_{T_r}$ of size $B_r$ each, divide $m$ into $T_r$ blocks $m_1, ..., m_{T_r}$ of size $B_r$ each.
5: **for** $1 \le j \le T_c$ **do**
6:     Load $\mathbf{K}_j, \mathbf{V}_j$ from HBM to on-chip SRAM.
7:     **for** $1 \le i \le T_r$ **do**
8:         **if** $M_{ij} \ne 0$ **then**
9:             Load $\mathbf{Q}_i, \mathbf{O}_i, \ell_i, m_i$ from HBM to on-chip SRAM.
10:            On chip, compute $\mathbf{S}_{ij} = \tau \mathbf{Q}_i \mathbf{K}_j^T \in \mathbb{R}^{B_r \times B_c}$.
11:            On chip, compute $\mathbf{S}_{ij}^{\text{masked}} = \text{MASK}(\mathbf{S}_{ij})$.
12:            On chip, compute $\tilde{m}_{ij} = \text{rowmax}(\mathbf{S}_{ij}^{\text{masked}}) \in \mathbb{R}^{B_r}, \tilde{\mathbf{P}}_{ij} = \exp(\mathbf{S}_{ij}^{\text{masked}} - \tilde{m}_{ij}) \in \mathbb{R}^{B_r \times B_c}$ (pointwise), $\tilde{\ell}_{ij} = \text{rowsum}(\tilde{\mathbf{P}}_{ij}) \in \mathbb{R}^{B_r}$.
13:            On chip, compute $m_i^{\text{new}} = \max(m_i, \tilde{m}_{ij}) \in \mathbb{R}^{B_r}, \ell_i^{\text{new}} = e^{m_i - m_i^{\text{new}}} \ell_i + e^{\tilde{m}_{ij} - m_i^{\text{new}}} \tilde{\ell}_{ij} \in \mathbb{R}^{B_r}$.
14:            On chip, compute $\tilde{\mathbf{P}}_{ij}^{\text{dropped}} = \text{dropout}(\tilde{\mathbf{P}}_{ij}, p_{\text{drop}})$.
15:            Write $\mathbf{O}_i \leftarrow \text{diag}(\ell_i^{\text{new}})^{-1}(\text{diag}(\ell_i) e^{m_i - m_i^{\text{new}}} \mathbf{O}_i + e^{\tilde{m}_{ij} - m_i^{\text{new}}} \tilde{\mathbf{P}}_{ij}^{\text{dropped}} \mathbf{V}_j)$ to HBM.
16:            Write $\ell_i \leftarrow \ell_i^{\text{new}}, m_i \leftarrow m_i^{\text{new}}$ to HBM.
17:         **end if**
18:     **end for**
19: **end for**
20: Return $\mathbf{O}, \ell, m, \mathcal{R}$.

---

We prove the IO-complexity of block-sparse FLASHATTENTION.

*Proof of Proposition 4.* The proof is very similar to the proof of Theorem 2. For the block-sparse case, notice that we only need to load blocks corresponding to nonzero blocks. As a result, the number of HBM accesses are scaled by $s$, the fraction of nonzero blocks in the block-sparsity mask. However, for small values of $s$, we would still need to write the result $\mathbf{O} \in \mathbb{R}^{N \times d}$. Therefore the number of HBM accesses is

$$\Theta\left(Nd + \frac{N^2 d^2}{M} s\right).$$

$\square$

## D.2  Potential Extensions

We discuss here a few potential extensions of the IO-aware approach to speed up deep learning training.

**Multi-GPU Attention.** Large language models are trained on hundreds or thousands of GPUs, and one typically splits the attention computation between 4-8 GPUs on the same node [80]. This introduces another level of memory hierarchy: beside GPU SRAM and GPU HBM, we also have the HBM of other GPUs. For very long sequences, the different GPUs on the same node can cooperate to compute attention by taking into account the asymmetry of different levels of memory hierarchy.

**Sparse MLP layers.** Typical dense MLP layers are compute-bound and not memory-bound. To improve their efficiency, MLP layers with sparse weight matrices can be used [18]. However, many sparse MLP layers are instead memory-bound, and their speedup is often not proportional to the sparsity. We believe that an IO-aware implementation can alleviate this issue and realize the benefits of sparsity. We are excited about future work in this direction, to reduce the computational requirement of large models and improve their wall-block runtime.

**Kernel machine learning.** Our approach in FLASHATTENTION relies on the fact that the $N \times N$ attention matrix is a function of a low-rank matrix $\mathbf{QK}^\top$ (of rank $d \ll N$). As a result, we can repeatedly load the inputs $\mathbf{Q}, \mathbf{K}$ and recompute the block of the attention matrix that we need, significantly reducing HBM access. As similar scenario happens in kernel machine learning: each element $K_{ij}$ of the $N \times N$ kernel matrix $\mathbf{K}$ is a function of two vectors of size $d \ll N$, as it measures the similarity between two datapoints $x_i$ and $x_j$. The KeOps library [8, 28] is a successful example of how reducing memory reads/writes can speed up kernel operations. We hope that this will motivate kernel methods that focus more on reducing IOs instead of just FLOPs.

# E  Full Experimental Results

## E.1  BERT

We train BERT-large following the training procedure and hyperparameters of the reference MLPerf 1.1 implementation. In particular, we use the LAMB optimizer with learning rate 3.75e-3, with batch size 448, trained for at most 7100 steps. The training is stopped once the validation accuracy (for masked language modeling) reaches the target 72.0%, and the wall-clock run-time is measured. We train with FP16 precision using Apex AMP (with O2 optimization level).

We compare our results with the reported training speed from Nvidia that was submitted to MLPerf 1.1 (Table 1).

We use the same train / validation data split provided by MLPerf 1.1 reference implementation. In particular, we evaluate on the same 10000 validation examples as the baseline from Nvidia.

We train the model on 8×A100-80GB GPUs. Each training run takes between 16 and 19 minutes, and we average the results of 10 runs.

We see a memory saving of $1.8 times$ (from 58GB to 32GB) for the same batch size.

In Table 7, we additionally compare against the commonly used Huggingface implementation. FLASHATTENTION is 3.2× faster than this implementation.

Table 7: Training time of BERT-large, starting from the same initialization provided by the MLPerf benchmark, to reach the target accuracy of 72.0% on masked language modeling. Averaged over 10 runs on 8×A100 GPUs.

| BERT Implementation | Training time (minutes) |
|---|---|
| Huggingface [91] | 55.6 ± 3.9 |
| Nvidia MLPerf 1.1 [63] | 20.0 ± 1.5 |
| FLASHATTENTION (ours) | **17.4** ± 1.4 |

## E.2  GPT-2

We use the standard implementations of GPT-2 [70] from Huggingface `transformers` library and from Nvidia's Megatron-LM repo. We follow the training recipe of the Megatron-LM repo.

We use an effective batch size of 512, and use gradient accumulation to fit into available GPU memory. We use the AdamW optimizer, with learning rate 6e-4 for GPT-2 small and 1.5e-4 for GPT-2 medium, and weight decay of 0.1. All models are trained with the same hyperparameters for 400K steps. We run all implementations with mixed-precision training (PyTorch AMP).

We use the Openwebtext dataset, with the GPT-2 BPE tokenizer. We randomly select 0.5% of the dataset as the validation set, with the rest being used as training set. This random selection of validation set is done once, and all models are evaluated on the same validation set.

We train the model on 8×A100-40GB GPUs, and we measure the wall-clock training time. Training GPT-2 small takes between 2.7-9.5 days, and training GPT-2 medium takes between 6.9-21.0 days (Table 2).

For GPT-2 small, we see a memory saving of $3.5 times$ (from 39GB to 11GB) for the same batch size of 16 (which means we could run FLASHATTENTION with 4× larger device batch size while keeping the global batch size of 512 the same).

In Fig. 4, we plot of the validation perplexity throughout training of GPT-2 small/medium, using either HuggingFace implementation or our FLASHATTENTION implementation. We see that FLASHATTENTION behaves the same as the baseline implementation and the validation perplexity curves of the two implementations almost lie on top of each other.

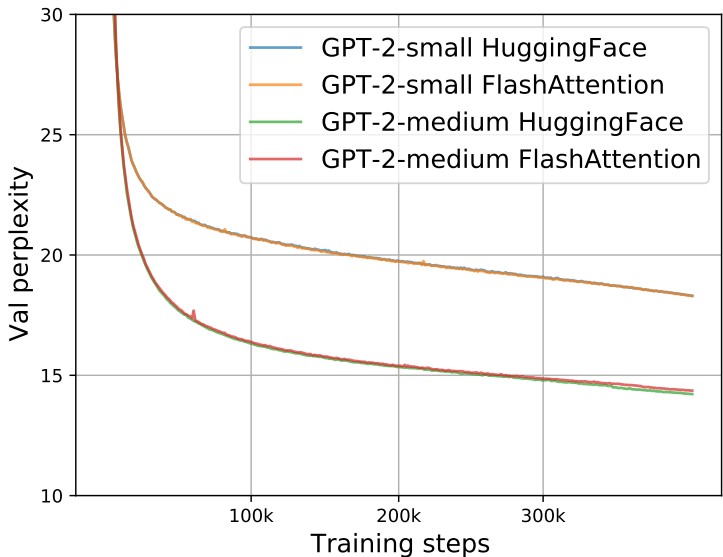

Figure 4: Validation perplexity of GPT-2 small/medium using two implementations. We confirm that FLASHATTENTION yields the same validation curves as the baseline implementation from HuggingFace.

We additionally compare the speedup of FlashAttention as we scale the number of GPUs from 1 to 8, for GPT-2 Small. All the training hyperparameters are kept the same. As we change the number of GPUs, we change the number of gradient accumulations to keep the global batch size the same (512). We use PyTorch DistributedDataParallel when there are more than 1 GPUs. Table 8 shows that the speedup is consistent, around 3.5-3.7×.

Table 8: Training speedup (in wallclock-time) of FlashAttention compared to Huggingface implementation on GPT-2 small as we vary the number of GPUs, measured on A100-SXM4-40GB GPUs. The speedup varies from 3.7× to 3.5×.

| FLASHATTENTION vs. Huggingface on GPT-2 Small | 1 GPU | 2 GPUs | 4 GPUs | 8 GPUs |
|---|---|---|---|---|
| Wallclock-time speedup | 3.7× | 3.6× | 3.6× | 3.5× |

**Long Document Classification.** For MIMIC-III and ECtHR, we follow the hyperparameters of Dai et al. [14].

### E.3  LRA details

We follow the hyperparameters from the Long-range arena paper [83], the Long-range arena repo (https://github.com/google-research/long-range-arena), and the Nyströmformer reproduction [94]. To be generous to the baseline methods, if we are unable to reproduce the performance of any baseline for any of the five tasks, we report the better performance from Tay et al. [83] or Xiong et al. [94] for that baseline on that task.

After hyperparameter tuning, almost all of the attention methods achieve similar accuracy on all of the five LRA tasks.

We run all methods with mixed-precision training, except for Performer (not stable with mixed precision) and Local Attention (implementation does not support FP16).

To calculate the overall wallclock-time speedup, we take the geometric mean of the wallclock-time speedup of each of the five tasks.

**Path-X** For Path-X and Path-256, we follow the hyperparameters from the PathFinder-32 experiments from the long-range arena paper[83]. For both, we first pretrain a model on Path-64.

We take the checkpoint after 200 epochs, upsample its positional embedding (we duplicate the positional embeddings gridwise in space), and fine-tune it on the downstream task for 200 epochs with one epoch of linear warmup, and cosine decay of the learning rate. For Path-X, we take the best performing checkpoint (according to val accuracy), and additionally fine-tune it for 200 epochs with the same warmup and learning rate (this adds roughly 4 points of accuracy to FLASHATTENTION for Path-X, but the model starts overfitting afterwards).

### E.4 Faster Vision Transformer with FLASHATTENTION on ImageNet

On the popular vision benchmark, ImageNet [22], we show that FLASHATTENTION can also speedup Vision Transformers (ViT) [26] by 1.5$times$, where the sequence length is 196 (patch size 16×16 for 224×224 images). For longer sequences, FLASHATTENTION yields up to 3.5× speedup.

We use the ViT-base implementation from the widely-used library `timm`, and replace the standard attention implementation with FLASHATTENTION. We follow the same training recipe as that of DeiT [85], which improves on the original training recipe of ViT. We measure accuracy and training time of both models (for 300 epochs) on 8×A100s. Table 9 shows that FLASHATTENTION achieves up 1.5× speed-up compared to standard attention.

Table 9: Training time of ViT-base on ImageNet for 300 epochs, on 8×A100 GPUs. Even with relatively small sequence length (196), FLASHATTENTION still yields 1.5x speedup.

| ViT-base implementation | ImageNet top-1 val accuracy | Training time (hours) |
|---|---|---|
| `timm` | 81.8% | 29.1 |
| FLASHATTENTION (ours) | 81.8% | **19.5** |

We also compare ViT-Large with smaller patch sizes (i.e., longer sequence lengths). Table 10 shows that FLASHATTENTION yields 3.5× speedup and saves up to 3.6$x$ memory, compared to standard attention.

Table 10: Forward + Backward time of ViT-Large on a batch of 224×224 images an A100 GPU. With longer sequence length, FLASHATTENTION yields 3.5x speedup and up to 3.6× memory saving.

| ViT-Large implementation | Sequence length | Batch size | Fwd + bwd time | Memory |
|---|---|---|---|---|
| ViT-Large (`timm`) patch size 8 | 784 | 32 | 1400ms | 36GB |
| ViT-Large (FLASHATTENTION) patch size 8 | 784 | 32 | **405ms** (3.5×) | **22GB** |
| ViT-Large (`timm`) patch size 4 | 3136 | 2 | 1200ms | 22GB |
| ViT-Large (FLASHATTENTION) patch size 4 | 3136 | 2 | **350ms** (3.4×) | **6GB** |

### E.5 Comparison with Automatic Fusion

We compare FlashAttention with automatic fusion methods: NVFuser from Pytorch 1.12 (newest version at the time of writing), AOT compiler from Functorch, and TVM [11]. For context, we also include the runtime of standard implementation from Pytorch and the more optimized implementation from Megatron-LM [80].

We benchmark for batch size 16, 32, and 64, sequence length 1024, 16 heads, head dimension 64, with key-padding mask (and no dropout). The runtime is measured on an A100-SXM4-40GB GPU. Table 11 shows that FLASHATTENTION is about 2-3× faster than these methods.

Table 11: Runtime (ms) of FLASHATTENTION compared to automatic fusion methods by sequence length, with key padding masking, measured on an A100-SXM4-40GB GPU. Batch size 16, 32, and 64, sequence length 1024, 16 heads, head dimension 64. FLASHATTENTION is 2-3× faster than these methods.

| Method | Batch size 16 | | | Batch size 32 | | | Batch size 64 | | |
|---|---|---|---|---|---|---|---|---|---|
| | Fwd | Bwd | Total | Fwd | Bwd | Total | Fwd | Bwd | Total |
| Pytorch eager mode | 4.1 | 5.0 | 9.1 | 8.1 | 9.5 | 17.6 | 16.1 | 19.0 | 35.1 |
| Pytorch JIT (NVFuser) | 2.8 | 4.8 | 7.6 | 5.5 | 9.5 | 15.0 | 11.0 | 18.7 | 29.7 |
| AOT compiler (Functorch) | 2.7 | 4.9 | 7.6 | 5.4 | 9.8 | 15.2 | 10.8 | 19.6 | 30.4 |
| TVM | 2.7 | 4.9 | 7.6 | 5.5 | 9.7 | 15.2 | 11.0 | 19.0 | 30.0 |
| Megatron-LM | 2.9 | 3.8 | 6.9 | 5.5 | 7.1 | 12.6 | 11.6 | 14.3 | 25.9 |
| FLASHATTENTION | **1.0** | **2.6** | **3.6** | **1.8** | **4.1** | **5.9** | **3.3** | **8.4** | **11.7** |

### E.6 Comparison with Apex FMHA

We compare our method/implementation with Apex FMHA (https://github.com/NVIDIA/apex/tree/master/apex/contrib/csrc/fmha).

Table 12: Runtime (ms) of FLASHATTENTION compared to FMHA by sequence length, with masking and dropout, measured on an A100-SXM4-40GB GPU. Batch size 64, 16 heads, head dimension 64 (i.e., BERT-large size).

| Attention Method | 128 | 256 | 512 |
|---|---|---|---|
| **Apex FMHA forward** | 0.10 | 0.29 | 1.14 |
| **FLASHATTENTION forward** | **0.08** | **0.22** | **0.81** |
| **Apex FMHA backward** | **0.17** | **0.52** | **1.81** |
| **FLASHATTENTION backward** | 0.20 | 0.53 | 2.00 |
| **Apex FMHA forward + backward** | **0.27** | 0.81 | 2.95 |
| **FLASHATTENTION forward + backward** | 0.28 | **0.75** | **2.81** |

When we started this project, Apex FMHA was the fastest implementation of attention (that we knew of), tailored for short sequences of length at most 512. In fact, almost all MLPerf submissions for BERT training benchmark running on Nvidia GPUs use FMHA for their model code, as of MLPerf 1.1 [60]. Since FMHA targets BERT models, it only supports head dimension 64, and only runs on A100 GPUs. FMHA fuses the attention computation dropout(softmax(MASK($\mathbf{QK}^\top$)))$\mathbf{V}$ into one CUDA kernel. In the forward pass, it stores the attention matrix softmax(MASK($\mathbf{QK}^T$)) to HBM to be used in gradient computation. As a result, it does not offer substantial memory saving (though for shorter sequences memory footprint is often not a primary concern).

We use FMHA code as a starting point, and apply two well-established techniques (tiling and recomputation) to deal with long sequences and to save memory as mentioned in Section 3. As a result, we can support much longer sequences (e.g., up to length 64K). We also support more head dimensions (16, 32, 64, 128) and broader GPU types (all Turing and Ampere GPUs at the time of writing).

In Table 12, we compare the performance of FLASHATTENTION and Apex FMHA for short sequences (as FMHA only supports sequence length at most 512). Generally FLASHATTENTION is slightly faster than FMHA in the forward pass and slightly slower than FMHA in the backward pass. This is because we do not store the attention matrix in the forward pass and recompute it in the backward pass. Compared to FMHA, the overall runtime of FLASHATTENTION is about 4% slower for sequence length 128, 8% faster for sequence length 256, and 5% faster for sequence length 512.

### E.7    Roofline analysis

In Fig. 5, we include a roofline analysis of the FLASHATTENTION forward pass, taken from Nvidia Nsight Compute (batch size 16, seqlen 512, 16 heads, head dimension 64) on an A100-SXM4-40GB GPU.

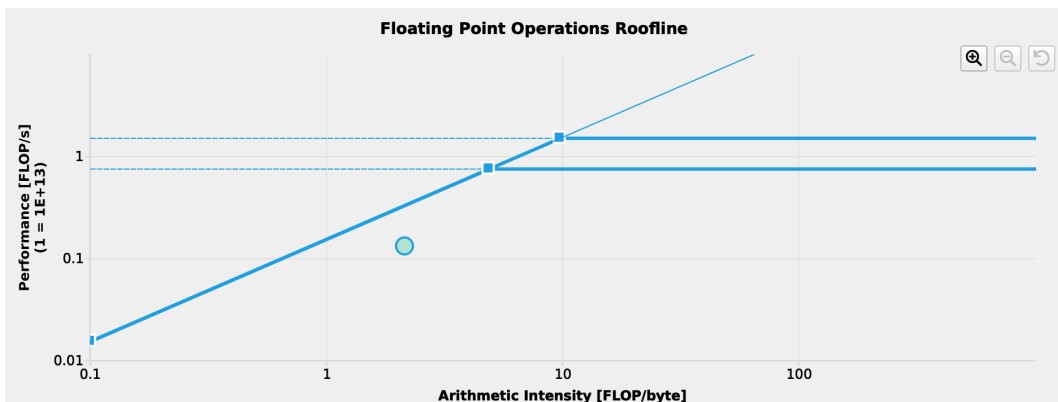

Figure 5: Roofline analysis of FLASHATTENTION forward pass. While FLASHATTENTION substantially speeds up attention, there is still some potential headroom to gain further speedup.

### E.8    Speedup On Different Hardware and Configurations

Speedup varies between different types of GPU types and generations depending on HBM bandwidth and SRAM size. In this section, we profile FLASHATTENTION speedup on different GPUs and configurations.

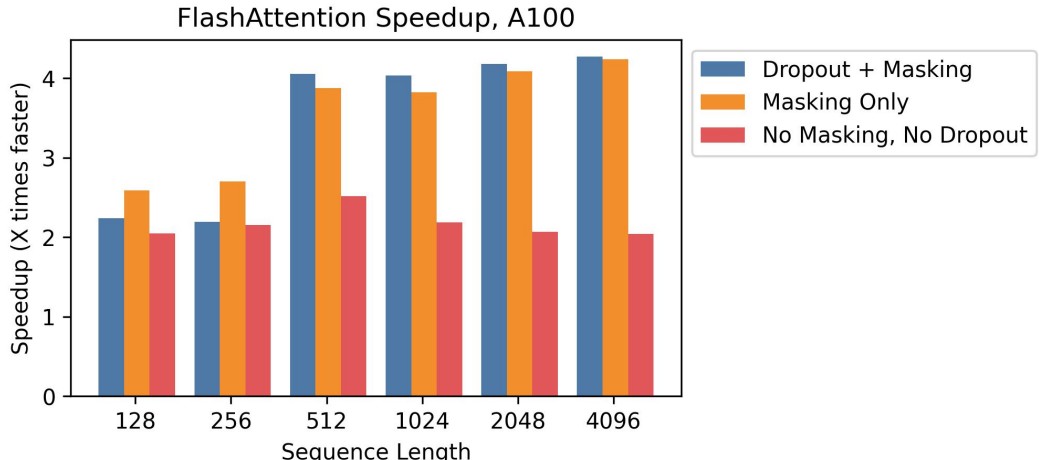

Figure 6: Speedup over standard PyTorch attention at different sequence lengths, on A100.

**A100** Figure 6 shows speedup on an A100 GPU with batch size 8, head dimension 64, and 12 attention heads, across different sequence lengths. We generally see 2-4× speedup, and we see more speedup when using dropout and masking due to kernel fusion.

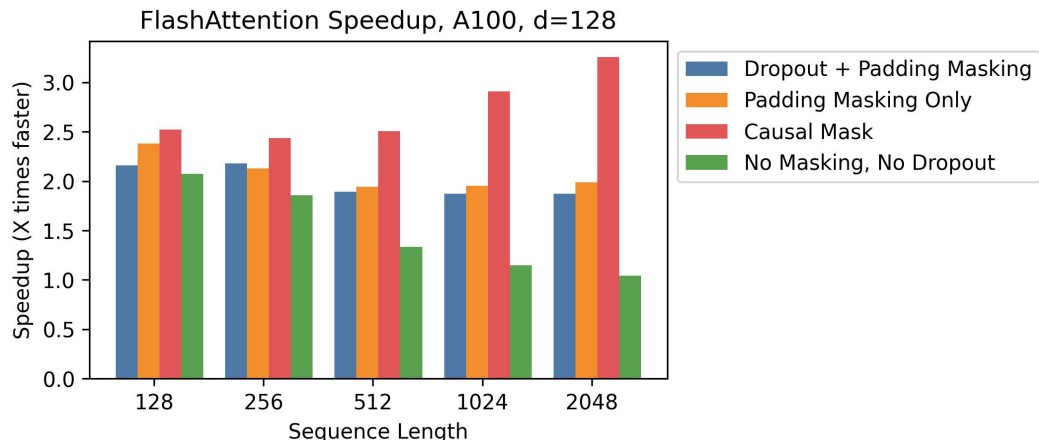

Figure 7: Speedup over standard PyTorch attention at different sequence lengths, on A100, with head dimension 128.

**A100, Head Dimension 128** Speedup also changes when we increase the head dimension. Each block requires more memory, so we need to use smaller block sizes to fit into SRAM. Figure 7 shows speedup with head dimension 128 on an A100 (batch size 16, 12 heads). We see less speedup overall—but we can still see significant speedup (up to 3×) with a causal mask, where half the blocks are masked out.

**RTX 3090** Figure 8 shows speedup on an RTX 3090 GPU. Here, we use batch size 12 with 12 attention heads. We observe slightly higher speedups on the RTX 3090 (between 2.5-4.5×), since the memory bandwidth on an RTX 3090 is lower than on an A100 (roughly 900 GB/s vs. 1.5 TB/s).

**T4** Figure 9 shows speedup on a T4 GPU. T4 SRAM is smaller than A100, so we need to make the block sizes smaller in FLASHATTENTION. As a result, we observe less speedup on T4, which matches the IO complexity analysis in Section 3.2. T4 GPUs are commonly used for inference, so we also report speedup on the forward pass only.

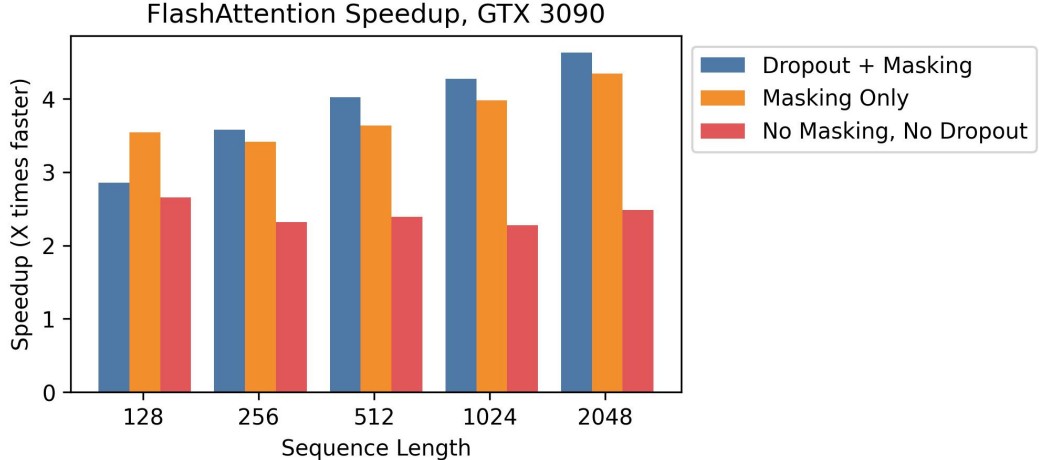

Figure 8: Speedup over standard PyTorch attention at different sequence lengths, on RTX 3090.

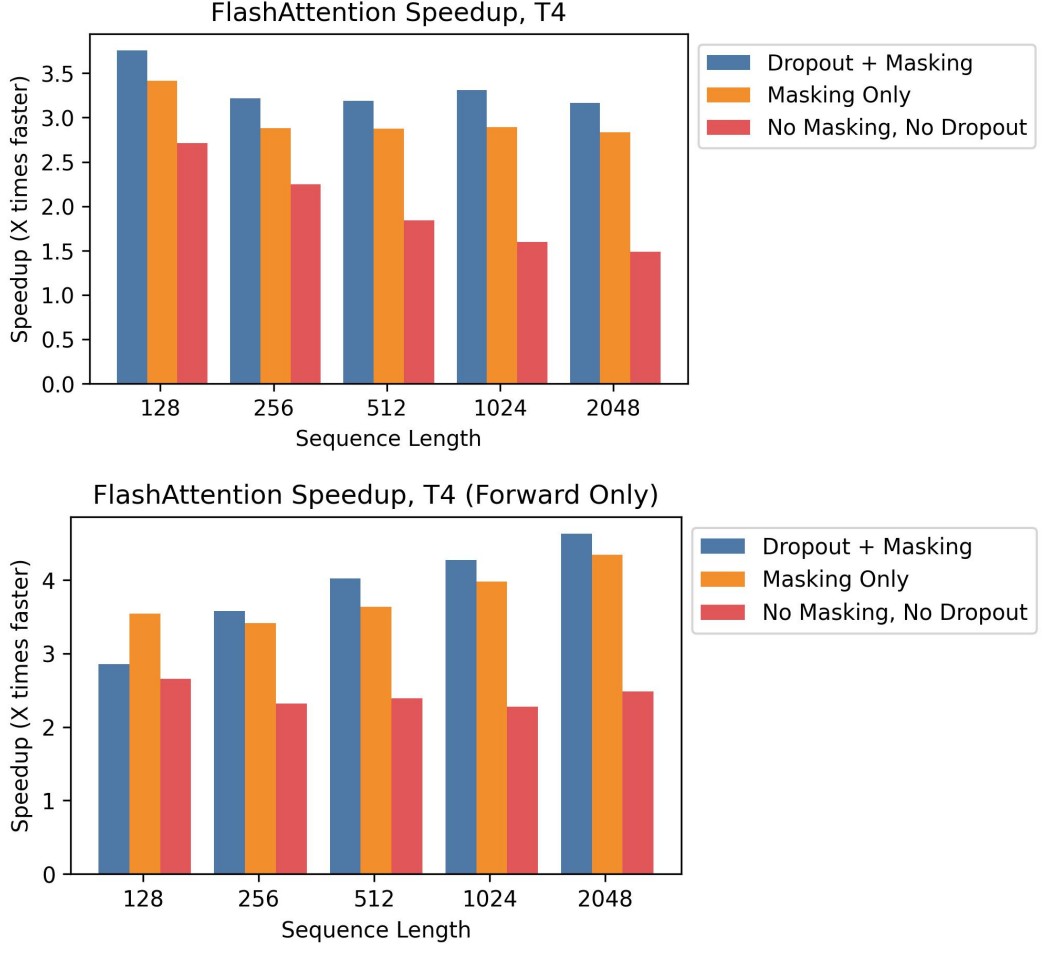

Figure 9: Speedup over standard PyTorch attention at different sequence lengths, on T4. **Top:** Combined forward pass + backward pass. **Bottom:** Forward pass only.

### E.9   Full Benchmarking Results

We report the full benchmarking results and experimental details on A100.

Table 13: Pointers to results tables.

| Dropout | Masking | Pass | Table |
|---------|---------|------|-------|
| Yes | Yes | Forward | Table 14 |
| Yes | Yes | Backward | Table 15 |
| Yes | Yes | Combined | Table 16 |
| No | Yes | Forward | Table 17 |
| No | Yes | Backward | Table 18 |
| No | Yes | Combined | Table 19 |
| Yes | No | Forward | Table 20 |
| Yes | No | Backward | Table 21 |
| Yes | No | Combined | Table 22 |
| No | No | Forward | Table 23 |
| No | No | Backward | Table 24 |
| No | No | Combined | Table 25 |
| No | No | Memory Usage (Combined) | Table 26 |

**Baselines** We compare against reference implementations for exact attention from PyTorch/HuggingFace and Megatron, approximate attention, and sparse attention. For approximate attention, we compare against reference implementations of Reformer [53], Local Attention [71], Linformer Attention [88], Smyrf [20], and LongShortFormer (LSFormer) [98]. For sparse attention, we compare against reference implementations of Block-Sparse Attention form OpenAI [12], Longformer[3], and BigBird Attention [96]. For the approximate and sparse attention, we use a compression ratio of 1/8, or a compressed sequence length of 256, whichever is smaller.

**Setup** We measure runtime and memory usage of the attention computation with 8 heads of dimension 64, and batch size 16 on a machine with one A100 GPU with 40 GB of GPU HBM. We vary sequence length in our experiments. We compute attention on random vectors for **Q**, **K**, and **V** (we do not measure the projection from the hidden layer). For dropout, we use dropout 0.1; for masking, we use a padding mask with uniformly-random mask lengths between the total sequence length and the total sequence length minus 20. To measure runtime, we take the average of 100 measurements of the attention call. We only measure memory footprint once, since it does not vary between runs.

We report timing results on the forward pass, backward pass, and combined forward + backward pass. We measure each method with and without dropout, masking, or both—except for Block Sparse, Longformer, and BigBird. These methods did not successfully run the backward pass with masking due to a bug in external libraries, so we measured them without masking to be generous. We use FP16 for all measurements, except for Local Attention, whose implementation only supports FP32.

For each baseline, we increase sequence length until it runs out of memory on the GPU, except for the following exceptions: The Megatron implementation does not support sequence lengths longer than 2048. Block-Sparse (OpenAI) does not support sequence lengths longer than 4096. Longformer and BigBird do not support sequence lengths longer than 8092.

We measure memory usage on the combined forward + backward pass, without dropout or masking.

**Results** Table 13 summarizes all the experimental configurations and contains pointers to the results tables.

Table 14: Forward pass runtime (ms) of various exact/approximate/sparse attention mechanisms by sequence length, **with dropout and masking**. Best in **bold**, second best underlined.

| Attention Method | 128 | 256 | 512 | 1024 | 2048 | 4096 | 8192 | 16384 | 32768 | 65536 |
|---|---|---|---|---|---|---|---|---|---|---|
| PyTorch Attention | 0.36 | 0.34 | 0.78 | 2.54 | 9.33 | 36.33 | - | - | - | - |
| Megatron | 0.40 | 0.40 | 1.10 | 3.65 | 16.19 | - | - | - | - | - |
| Reformer | 2.03 | 3.15 | 5.67 | 11.02 | 22.59 | 46.14 | 97.38 | 212.13 | - | - |
| Local Attention | 0.83 | 0.86 | 1.01 | 2.20 | 7.13 | 14.32 | 28.60 | 57.79 | 117.67 | - |
| Linformer | 0.67 | 0.52 | 0.69 | 0.71 | 1.65 | 3.18 | 6.15 | 12.16 | 24.17 | 52.39 |
| Smyrf | 2.27 | 2.34 | 3.91 | 7.44 | 14.71 | 29.22 | 58.27 | 116.41 | - | - |
| LSformer | 1.18 | 1.27 | 1.34 | 3.38 | 11.40 | 22.55 | 44.95 | 89.76 | 179.66 | - |
| Block Sparse | 1.12 | 1.11 | 2.13 | 2.77 | 6.95 | 20.91 | - | - | - | - |
| Longformer | 1.22 | 1.14 | 1.08 | 1.95 | 5.72 | 12.98 | - | - | - | - |
| BigBird | 1.13 | 1.12 | 1.12 | 1.77 | 6.03 | 13.68 | - | - | - | - |
| FLASHATTENTION | **0.04** | 0.06 | 0.21 | 0.82 | 2.85 | 10.41 | 41.74 | 167.19 | 670.76 | 2682.35 |
| Block-Sparse FLASHATTENTION | 0.06 | **0.06** | **0.06** | **0.12** | **0.44** | **0.86** | **1.70** | **3.29** | **6.55** | **13.34** |

Table 15: Backward pass runtime (ms) of various exact/approximate/sparse attention mechanisms by sequence length, **with dropout and masking**. Best in **bold**, second best underlined.

| Attention Method | 128 | 256 | 512 | 1024 | 2048 | 4096 | 8192 | 16384 | 32768 | 65536 |
|---|---|---|---|---|---|---|---|---|---|---|
| PyTorch Attention | 0.37 | 0.49 | 1.66 | 5.81 | 22.32 | 87.67 | - | - | - | - |
| Megatron | 0.35 | 0.32 | 0.77 | 2.42 | 8.43 | - | - | - | - | - |
| Reformer | 2.37 | 4.59 | 8.91 | 17.68 | 35.13 | 70.05 | 140.01 | - | - | - |
| Local Attention | 0.55 | 0.62 | 1.49 | 4.03 | 13.78 | 27.61 | 55.20 | 110.27 | 221.40 | - |
| Linformer | 0.89 | 0.80 | 0.81 | 0.93 | 2.48 | 4.75 | 9.29 | 18.27 | 36.53 | - |
| Smyrf | 1.41 | 2.83 | 5.43 | 10.72 | 21.25 | 42.31 | 84.48 | 168.95 | - | - |
| LSformer | 1.75 | 1.76 | 3.01 | 7.50 | 20.07 | 39.08 | 76.39 | 150.82 | - | - |
| Block Sparse | 1.29 | 1.28 | 2.18 | 3.04 | 7.27 | 21.16 | - | - | - | - |
| Longformer | 1.27 | 1.31 | 1.29 | 2.04 | 5.24 | 10.74 | 25.95 | - | - | - |
| BigBird | 1.33 | 1.28 | 1.32 | 1.81 | 5.55 | 11.44 | 27.45 | - | - | - |
| FLASHATTENTION | **0.30** | **0.26** | 0.68 | 2.02 | 6.84 | 26.89 | 105.70 | 418.96 | 1666.89 | 6660.44 |
| Block-Sparse FLASHATTENTION | **0.30** | 0.27 | **0.29** | **0.59** | **1.50** | **2.94** | **5.82** | **11.85** | **23.98** | **47.61** |

Table 16: Forward pass + backward pass runtime (ms) of various exact/approximate/sparse attention mechanisms by sequence length, **with dropout and masking**. Best in **bold**, second best underlined.

| Attention Method | 128 | 256 | 512 | 1024 | 2048 | 4096 | 8192 | 16384 | 32768 | 65536 |
|---|---|---|---|---|---|---|---|---|---|---|
| PyTorch Attention | 0.84 | 0.86 | 2.35 | 8.29 | 31.75 | 124.19 | - | - | - | - |
| Megatron | 0.87 | 0.89 | 1.33 | 4.21 | 16.50 | - | - | - | - | - |
| Reformer | 4.30 | 7.76 | 14.60 | 28.74 | 57.79 | 116.34 | 237.57 | - | - | - |
| Local Attention | 1.40 | 1.60 | 2.06 | 6.06 | 20.94 | 42.01 | 84.08 | 168.48 | 339.45 | - |
| Linformer | 1.57 | 1.49 | 1.55 | 1.60 | 4.19 | 8.04 | 15.71 | 30.92 | 61.47 | - |
| Smyrf | 3.41 | 5.08 | 9.35 | 18.18 | 36.03 | 71.68 | 143.04 | 285.87 | - | - |
| LSformer | 3.08 | 3.10 | 4.26 | 10.90 | 31.59 | 61.72 | 121.51 | 241.18 | - | - |
| Block Sparse | 2.54 | 2.52 | 3.71 | 5.44 | 13.29 | 39.19 | - | - | - | - |
| Longformer | 2.47 | 2.49 | 2.51 | 3.10 | 10.39 | 22.49 | 60.44 | - | - | - |
| BigBird | 2.51 | 2.49 | 2.52 | 3.40 | 10.97 | 23.89 | 63.28 | - | - | - |
| FLASHATTENTION | **0.43** | **0.41** | 0.95 | 2.55 | 9.56 | 37.49 | 147.75 | 586.61 | 2339.11 | 9341.30 |
| Block-Sparse FLASHATTENTION | 0.44 | 0.44 | **0.45** | **0.89** | **1.95** | **4.12** | **7.64** | **16.60** | **32.73** | **64.11** |

Table 17: Forward pass runtime (ms) of various exact/approximate/sparse attention mechanisms by sequence length, **with masking**. Best in **bold**, second best underlined.

| Attention Method | 128 | 256 | 512 | 1024 | 2048 | 4096 | 8192 | 16384 | 32768 | 65536 |
|---|---|---|---|---|---|---|---|---|---|---|
| PyTorch Attention | 0.30 | 0.30 | 0.63 | 1.93 | 7.08 | 27.45 | 112.90 | - | - | - |
| Megatron | 0.45 | 0.41 | 0.43 | 1.52 | 5.80 | - | - | - | - | - |
| Reformer | 1.87 | 3.00 | 5.37 | 10.43 | 21.40 | 43.83 | 92.80 | 203.24 | - | - |
| Local Attention | 0.70 | 0.81 | 1.02 | 2.09 | 6.64 | 13.34 | 26.77 | 54.02 | 110.11 | - |
| Linformer | 0.63 | 0.50 | 0.67 | 0.65 | 1.36 | 2.60 | 5.04 | 9.92 | 19.69 | 43.47 |
| Smyrf | 2.38 | 2.32 | 3.76 | 7.16 | 14.14 | 28.09 | 55.98 | 111.73 | - | - |
| LSformer | 1.22 | 1.29 | 1.44 | 3.28 | 10.99 | 21.72 | 43.29 | 86.32 | 172.76 | - |
| Block Sparse | 0.96 | 1.04 | 1.66 | 2.16 | 5.41 | 16.15 | - | - | - | - |
| Longformer | 0.99 | 0.98 | 0.99 | 1.56 | 4.79 | 11.07 | 32.98 | - | - | - |
| BigBird | 0.96 | 1.02 | 1.02 | 1.48 | 5.05 | 11.59 | 34.16 | - | - | - |
| FLASHATTENTION | **0.03** | **0.04** | 0.17 | 0.68 | 2.28 | 8.40 | 33.55 | 134.14 | 537.50 | 2150.88 |
| Block-Sparse FLASHATTENTION | 0.05 | **0.04** | **0.05** | **0.11** | **0.35** | **0.68** | **1.33** | **2.54** | **5.34** | **10.73** |

Table 18: Backward pass runtime (ms) of various exact/approximate/sparse attention mechanisms by sequence length, **with masking**. Best in **bold**, second best underlined.

| Attention Method | 128 | 256 | 512 | 1024 | 2048 | 4096 | 8192 | 16384 | 32768 | 65536 |
|---|---|---|---|---|---|---|---|---|---|---|
| **PyTorch Attention** | 0.44 | 0.46 | 1.53 | 5.33 | 20.34 | 79.87 | - | - | - | - |
| **Megatron** | 0.29 | 0.31 | 0.65 | 1.95 | 6.49 | - | - | - | - | - |
| **Reformer** | 2.31 | 4.47 | 8.68 | 17.20 | 34.14 | 68.09 | 136.02 | - | - | - |
| **Local Attention** | 0.51 | 0.62 | 1.30 | 3.81 | 13.33 | 26.72 | 53.41 | 106.82 | 214.15 | - |
| **Linformer** | 0.76 | 0.81 | 0.94 | 0.87 | 2.24 | 4.25 | 8.35 | 16.38 | 32.67 | 72.11 |
| **Smyrf** | 1.34 | 2.77 | 5.30 | 10.46 | 20.73 | 41.27 | 82.41 | 164.86 | - | - |
| **LSformer** | 1.66 | 1.61 | 3.09 | 7.42 | 19.68 | 38.35 | 74.92 | 147.86 | - | - |
| **Block Sparse** | 1.24 | 1.25 | 2.04 | 2.91 | 6.78 | 19.67 | - | - | - | - |
| **Longformer** | 1.27 | 1.23 | 1.24 | 1.85 | 4.99 | 10.21 | 24.89 | - | - | - |
| **BigBird** | 1.43 | 1.50 | 1.44 | 1.69 | 5.25 | 10.86 | 26.26 | - | - | - |
| **FLASHATTENTION** | **0.21** | **0.22** | 0.62 | 1.84 | 5.77 | 22.25 | 86.21 | 338.91 | 1343.91 | 5361.09 |
| **Block-Sparse FLASHATTENTION** | 0.22 | 0.22 | **0.26** | **0.57** | **1.55** | **3.13** | **5.98** | **12.21** | **23.49** | **47.85** |

Table 19: Forward pass + backward pass runtime (ms) of various exact/approximate/sparse attention mechanisms by sequence length, **with masking**. Best in **bold**, second best underlined.

| Attention Method | 128 | 256 | 512 | 1024 | 2048 | 4096 | 8192 | 16384 | 32768 | 65536 |
|---|---|---|---|---|---|---|---|---|---|---|
| **PyTorch Attention** | 0.80 | 0.81 | 2.08 | 7.23 | 27.51 | 107.58 | - | - | - | - |
| **Megatron** | 0.81 | 0.83 | 1.09 | 3.36 | 12.39 | - | - | - | - | - |
| **Reformer** | 4.16 | 7.46 | 14.06 | 27.68 | 55.66 | 112.15 | 229.37 | - | - | - |
| **Local Attention** | 1.39 | 1.68 | 2.08 | 5.83 | 20.04 | 40.16 | 80.44 | 161.35 | 325.11 | - |
| **Linformer** | 1.51 | 1.42 | 1.56 | 1.67 | 3.67 | 6.99 | 13.63 | 26.77 | 53.36 | 117.56 |
| **Smyrf** | 3.38 | 4.93 | 9.07 | 17.66 | 34.94 | 69.55 | 138.72 | 277.41 | - | - |
| **LSformer** | 3.08 | 3.10 | 4.26 | 10.90 | 31.59 | 61.72 | 121.51 | 241.18 | - | - |
| **Block Sparse** | 2.39 | 2.40 | 3.31 | 5.02 | 12.25 | 35.94 | - | - | - | - |
| **Longformer** | 2.36 | 2.34 | 2.38 | 2.94 | 9.83 | 21.35 | 58.12 | - | - | - |
| **BigBird** | 2.35 | 2.35 | 2.37 | 3.25 | 10.36 | 22.57 | 60.63 | - | - | - |
| **FLASHATTENTION** | **0.32** | **0.30** | 0.83 | 2.37 | 7.95 | 30.77 | 119.98 | 473.65 | 1883.43 | 7513.01 |
| **Block-Sparse FLASHATTENTION** | 0.34 | 0.34 | **0.36** | **0.69** | **1.85** | **3.89** | **7.16** | **14.85** | **30.46** | **60.03** |

Table 20: Forward pass runtime (ms) of various exact/approximate/sparse attention mechanisms by sequence length, **with dropout**. Best in **bold**, second best underlined.

| Attention Method | 128 | 256 | 512 | 1024 | 2048 | 4096 | 8192 | 16384 | 32768 | 65536 |
|---|---|---|---|---|---|---|---|---|---|---|
| **PyTorch Attention** | 0.26 | 0.24 | 0.57 | 1.80 | 6.56 | 25.34 | - | - | - | - |
| **Megatron** | 0.27 | 0.27 | 0.56 | 1.88 | 6.56 | - | - | - | - | - |
| **Reformer** | 1.83 | 2.96 | 5.31 | 10.33 | 21.19 | 43.42 | 91.96 | 201.34 | - | - |
| **Local Attention** | 0.51 | 0.60 | 0.78 | 2.01 | 6.23 | 12.52 | 25.07 | 50.50 | 102.18 | - |
| **Linformer** | 0.47 | 0.37 | 0.49 | **0.52** | 1.37 | 2.65 | 5.12 | 10.13 | 20.25 | 44.16 |
| **Smyrf** | 2.12 | 2.01 | 3.15 | 5.97 | 11.83 | 23.36 | 46.48 | 92.72 | - | - |
| **LSformer** | 1.28 | 1.33 | 1.51 | 3.39 | 11.40 | 22.54 | 44.96 | 89.85 | 179.73 | - |
| **Block Sparse** | 1.03 | 1.00 | 1.72 | 2.39 | 5.96 | 17.88 | - | - | - | - |
| **Longformer** | 1.02 | 1.03 | 1.03 | 1.73 | 5.10 | 11.63 | 34.22 | - | - | - |
| **BigBird** | 0.99 | 1.03 | 1.01 | 1.58 | 5.36 | 12.27 | 35.56 | - | - | - |
| **FLASHATTENTION** | **0.10** | **0.10** | 0.22 | 0.83 | 2.81 | 10.38 | 41.63 | 167.01 | 668.74 | 2678.11 |
| **Block-Sparse FLASHATTENTION** | 0.54 | 0.51 | 0.68 | 0.61 | **0.67** | **1.10** | **1.89** | **3.71** | **7.18** | **14.41** |

Table 21: Backward pass runtime (ms) of various exact/approximate/sparse attention mechanisms by sequence length, **with dropout**. Best in **bold**, second best underlined.

| Attention Method | 128 | 256 | 512 | 1024 | 2048 | 4096 | 8192 | 16384 | 32768 | 65536 |
|---|---|---|---|---|---|---|---|---|---|---|
| **PyTorch Attention** | 0.44 | 0.35 | 0.90 | 2.94 | 10.77 | 41.67 | - | - | - | - |
| **Megatron** | 0.28 | 0.33 | 0.92 | 2.94 | 10.80 | - | - | - | - | - |
| **Reformer** | 2.24 | 4.34 | 8.39 | 16.62 | 33.02 | 65.77 | 131.52 | - | - | - |
| **Local Attention** | 0.51 | 0.58 | 1.41 | 3.71 | 12.96 | 25.98 | 51.94 | 103.72 | 207.78 | - |
| **Linformer** | 0.84 | 0.74 | 0.79 | 0.85 | 2.28 | 4.37 | 8.66 | 17.02 | 33.78 | - |
| **Smyrf** | 1.27 | 2.56 | 4.90 | 9.66 | 19.16 | 38.13 | 76.17 | 152.39 | - | - |
| **LSformer** | 1.67 | 1.77 | 3.03 | 7.52 | 20.10 | 39.13 | 76.35 | 150.83 | - | - |
| **Block Sparse** | 1.27 | 1.36 | 2.15 | 3.04 | 7.27 | 21.18 | - | - | - | - |
| **Longformer** | 1.28 | 1.34 | 1.38 | 1.98 | 5.24 | 10.74 | 25.95 | - | - | - |
| **BigBird** | 1.48 | 1.47 | 1.50 | 1.81 | 5.57 | 11.38 | 27.43 | - | - | - |
| **FLASHATTENTION** | **0.15** | 0.18 | 0.58 | 1.86 | 6.50 | 26.21 | 104.27 | 416.10 | 1661.92 | 6643.01 |
| **Block-Sparse FLASHATTENTION** | 0.17 | **0.17** | **0.17** | **0.40** | **1.10** | **2.04** | **4.43** | **9.33** | **18.28** | **37.31** |

Table 22: Forward pass + backward pass runtime (ms) of various exact/approximate/sparse attention mechanisms by sequence length, **with dropout**. Best in **bold**, second best underlined.

| Attention Method | 128 | 256 | 512 | 1024 | 2048 | 4096 | 8192 | 16384 | 32768 | 65536 |
|---|---|---|---|---|---|---|---|---|---|---|
| PyTorch Attention | 0.66 | 0.67 | 1.43 | 4.82 | 17.47 | 67.29 | - | - | - | - |
| Megatron | 0.88 | 0.90 | 1.49 | 4.73 | 17.41 | - | - | - | - | - |
| Reformer | 4.06 | 7.28 | 13.68 | 26.98 | 54.27 | 109.39 | 223.80 | - | - | - |
| Local Attention | 1.09 | 1.40 | 1.99 | 5.61 | 19.23 | 38.62 | 77.30 | 154.63 | 311.12 | - |
| Linformer | 1.31 | 1.21 | 1.30 | 1.39 | 3.73 | 7.15 | 14.05 | 27.69 | 55.00 | - |
| Smyrf | 3.00 | 4.37 | 8.05 | 15.66 | 31.04 | 61.64 | 123.04 | 245.65 | - | - |
| LSformer | 3.07 | 3.17 | 4.31 | 10.89 | 31.54 | 61.78 | 121.56 | 240.94 | - | - |
| Block Sparse | 2.54 | 2.52 | 3.71 | 5.44 | 13.29 | 39.19 | - | - | - | - |
| Longformer | 2.47 | 2.49 | 2.51 | 3.10 | 10.39 | 22.49 | 60.44 | - | - | - |
| BigBird | 2.51 | 2.49 | 2.52 | 3.40 | 10.97 | 23.89 | 63.28 | - | - | - |
| FLASHATTENTION | **0.35** | **0.36** | **0.80** | 2.52 | 9.16 | 36.70 | 146.13 | 583.45 | 2332.01 | 9323.63 |
| Block-Sparse FLASHATTENTION | 0.91 | 0.83 | 0.94 | **0.92** | **1.83** | **3.50** | **7.02** | **13.56** | **26.71** | **53.92** |

Table 23: Forward pass runtime (ms) of various exact/approximate/sparse attention mechanisms by sequence length. Best in **bold**, second best underlined.

| Attention Method | 128 | 256 | 512 | 1024 | 2048 | 4096 | 8192 | 16384 | 32768 | 65536 |
|---|---|---|---|---|---|---|---|---|---|---|
| PyTorch Attention | 0.21 | 0.22 | 0.43 | 1.27 | 4.32 | 16.47 | 67.77 | - | - | - |
| Megatron | 0.24 | 0.26 | 0.42 | 1.33 | 4.28 | - | - | - | - | - |
| Reformer | 1.77 | 2.82 | 5.01 | 9.74 | 20.03 | 41.11 | 87.39 | 192.40 | - | - |
| Local Attention | 0.48 | 0.57 | 0.80 | 1.90 | 5.76 | 11.56 | 23.13 | 46.65 | 94.74 | - |
| Linformer | 0.46 | 0.36 | 0.45 | **0.50** | 1.09 | 2.09 | 4.01 | 7.90 | 15.70 | 35.40 |
| Smyrf | 1.94 | 1.96 | 3.01 | 5.69 | 11.26 | 22.23 | 44.21 | 88.22 | - | - |
| LSformer | 1.21 | 1.34 | 1.34 | 3.31 | 11.01 | 21.71 | 43.27 | 86.32 | 172.85 | - |
| Block Sparse | 0.96 | 1.04 | 1.66 | 2.16 | 5.41 | 16.15 | - | - | - | - |
| Longformer | 0.99 | 0.98 | 0.99 | 1.56 | 4.79 | 11.07 | 32.98 | - | - | - |
| BigBird | 0.96 | 1.02 | 1.02 | 1.48 | 5.05 | 11.59 | 34.16 | - | - | - |
| FLASHATTENTION | **0.08** | **0.09** | **0.18** | 0.68 | 2.40 | 8.42 | 33.54 | 134.03 | 535.95 | 2147.05 |
| Block-Sparse FLASHATTENTION | 0.56 | 0.52 | 0.63 | 0.65 | **0.61** | **0.96** | **1.69** | **3.02** | **5.69** | **11.77** |

Table 24: Backward pass runtime (ms) of various exact/approximate/sparse attention mechanisms by sequence length. Best in **bold**, second best underlined.

| Attention Method | 128 | 256 | 512 | 1024 | 2048 | 4096 | 8192 | 16384 | 32768 | 65536 |
|---|---|---|---|---|---|---|---|---|---|---|
| PyTorch Attention | 0.26 | 0.29 | 0.78 | 2.44 | 8.82 | 33.87 | - | - | - | - |
| Megatron | 0.29 | 0.30 | 0.80 | 2.59 | 8.86 | - | - | - | - | - |
| Reformer | 2.18 | 4.21 | 8.14 | 16.12 | 32.02 | 63.84 | 127.60 | - | - | - |
| Local Attention | 0.51 | 0.64 | 1.28 | 3.60 | 12.52 | 25.08 | 50.22 | 100.23 | 200.66 | - |
| Linformer | 0.69 | 0.76 | 0.69 | 0.80 | 2.04 | 3.88 | 7.67 | 15.04 | 30.11 | 63.15 |
| Smyrf | 1.24 | 2.49 | 4.77 | 9.42 | 18.65 | 37.12 | 74.15 | 148.35 | - | - |
| LSformer | 1.68 | 1.61 | 3.02 | 7.40 | 19.72 | 38.27 | 74.89 | 147.99 | - | - |
| Block Sparse | 1.24 | 1.25 | 2.04 | 2.91 | 6.78 | 19.67 | - | - | - | - |
| Longformer | 1.27 | 1.23 | 1.24 | 1.85 | 4.99 | 10.21 | 24.89 | - | - | - |
| BigBird | 1.43 | 1.50 | 1.44 | 1.69 | 5.25 | 10.86 | 26.26 | - | - | - |
| FLASHATTENTION | **0.11** | 0.16 | 0.52 | 1.62 | 5.45 | 21.57 | 84.75 | 336.00 | 1338.56 | 5343.19 |
| Block-Sparse FLASHATTENTION | 0.11 | **0.12** | **0.16** | **0.38** | **1.20** | **2.34** | **4.69** | **9.10** | **18.74** | **37.04** |

Table 25: Forward pass + backward pass runtime (ms) of various exact/approximate/sparse attention mechanisms by sequence length. Best in **bold**, second best underlined.

| Attention Method | 128 | 256 | 512 | 1024 | 2048 | 4096 | 8192 | 16384 | 32768 | 65536 |
|---|---|---|---|---|---|---|---|---|---|---|
| PyTorch Attention | 0.67 | 0.70 | 1.18 | 3.67 | 13.22 | 50.44 | - | - | - | - |
| Megatron | 0.74 | 0.65 | 1.23 | 3.80 | 13.21 | - | - | - | - | - |
| Reformer | 3.93 | 7.01 | 13.15 | 25.89 | 52.09 | 105.00 | 215.13 | - | - | - |
| Local Attention | 1.09 | 1.27 | 1.99 | 5.38 | 18.32 | 36.77 | 73.67 | 147.29 | 296.35 | - |
| Linformer | 1.31 | 1.25 | 1.30 | 1.29 | 3.20 | 6.10 | 11.93 | 23.39 | 46.72 | 100.52 |
| Smyrf | 2.98 | 4.23 | 7.78 | 15.12 | 29.96 | 59.45 | 118.60 | 237.02 | - | - |
| LSformer | 3.03 | 3.05 | 4.26 | 10.70 | 30.77 | 60.15 | 118.33 | 234.94 | - | - |
| Block Sparse | 2.39 | 2.40 | 3.31 | 5.02 | 12.25 | 35.94 | - | - | - | - |
| Longformer | 2.36 | 2.34 | 2.38 | 2.94 | 9.83 | 21.35 | 58.12 | - | - | - |
| BigBird | 2.35 | 2.35 | 2.37 | 3.25 | 10.36 | 22.57 | 60.63 | - | - | - |
| FLASHATTENTION | **0.31** | **0.31** | **0.73** | 2.29 | 7.64 | 30.09 | 118.50 | 470.51 | 1876.08 | 7492.85 |
| Block-Sparse FLASHATTENTION | 0.74 | 0.77 | 0.82 | **0.88** | **1.71** | **3.21** | **6.56** | **12.60** | **24.93** | **50.39** |

Table 26: Memory usage (MB) of various exact/approximate/sparse attention mechanisms by sequence length. Best in **bold**, second best underlined.

| Attention Method | 128 | 256 | 512 | 1024 | 2048 | 4096 | 8192 | 16384 | 32768 | 65536 |
|---|---|---|---|---|---|---|---|---|---|---|
| **PyTorch Attention** | 36 | 104 | 336 | 1184 | 4416 | 17024 | - | - | - | - |
| **Megatron** | 36 | 104 | 336 | 1184 | 4416 | - | - | - | - | - |
| **Reformer** | 377 | 754 | 1508 | 3016 | 6033 | 12067 | 24134 | - | - | - |
| **Local Attention** | 53 | 110 | 232 | 592 | 1696 | 3392 | 6784 | 13568 | 27136 | - |
| **Linformer** | 25 | 52 | 114 | 287 | 832 | 1652 | 3292 | 6572 | 13132 | 26252 |
| **Smyrf** | 217 | 434 | 868 | 1737 | 3474 | 6947 | 13894 | 27788 | - | - |
| **LSformer** | 72 | 152 | 333 | 796 | 2540 | 5068 | 10125 | 20240 | - | - |
| **Block Sparse** | 33 | 82 | 228 | 408 | 910 | 2401 | - | - | - | - |
| **Longformer** | 30 | 61 | 124 | 277 | 681 | 1370 | 2748 | - | - | - |
| **BigBird** | 33 | 66 | 131 | 294 | 708 | 1431 | 2872 | - | - | - |
| **FLASHATTENTION** | **22** | **44** | **104** | **209** | **418** | **836** | **1672** | **3344** | **6688** | **13376** |
| **Block-Sparse FLASHATTENTION** | 22 | 44 | 104 | 209 | 418 | 836 | 1672 | 3344 | 6690 | 13384 |