# OpenReview forum: "FlashAttention: Fast and Memory-Efficient Exact Attention with IO-Awareness"
_NeurIPS.cc/2022/Conference — NeurIPS 2022 Accept_

### Official Review · Reviewer_5Whx · 2022-06-25

**Rating:** 7
**Confidence:** 4
**Soundness:** 3 good
**Presentation:** 3 good
**Contribution:** 3 good

**Summary:**

The paper proposes an algorithm to compute exact attention that reduces memory data transfers.
They tile computation of softmax, which is a major bottleneck in running transformers.
And they recompute attention matrix to avoid storing it in memory.
They also propose block-sparse attention in which they eliminate zero blocks in model with high sparsity.
This improves training time and produces more accurate models because it can model longer sequence.
They provide IO-complexity for the algorithm.
Previous methods that change the attention module and use approximate function of softmax.
The proposed method was evaluated on transformer models (e.g: BERT, GPT2). They show improved training throughput and ability to use longer sequence lengths and showcase this in Path-X and Path-256 challenges.


**Questions:**

Some suggestions and questions that could improve the paper are:

Remains to be seem if the method is relevant and if can be applied to mobile variation of transformers (mobile-ViT) and Vision transformers.

The effectiveness of the method is bounded by hardware constraints (SRAM size). It would interesting to see a roofline plot to demonstrate the compute-bound and memory access trade-off with and without flash-attention.

Evaluation or comment on other memory technologies other than HBM and GPUs to expand coverage of this work

Compilers (e.g: TVM) have mechanisms for tiling and adding passes. It may be interesting to see if other optimizations in TVM could be applied here and further improve Flashattention

**Limitations:**

Yes. the paper appropriately discuss limitations and social impact

**Strengths And Weaknesses:**

Pros:
The paper tackles an important problem and demonstrates algorithm optimizations to improve training runtime and long-range accuracy of relevant transformer models.

The claims are supported by evaluations

Overall the paper is clear

Cons:
The techniques: tiling and recomputation are known algorithms that is applied into the new transformers workload.

---

> ### Author Response · Authors · 2022-08-02
> **Response to Reviewer 5Whx**
>
> Thank you for your comments.
>
> **Q. Techniques used are known.**
> We discuss this in the common response.
>
> **Q: Vision transformers.**
> FlashAttention can also speed up Vision Transformer (1.5x for seqlen 196, 3.4x for seqlen 3136) and save memory (3.6x). We have updated the paper (Appendix E.4) to include an experiment with ViT-base on ImageNet: we obtain 1.5x speedup even when the seqlen is quite small (196) for ViT-base with patch size 16x16. Our benchmark on ViT-large with patch size 4x4 (seqlen 3136) yields 3.4x speedup and 3.6x memory saving. We are excited about this application with high-resolution images. Thank you for this great suggestion.
>
> **Q: Roofline plot**
> We have added a roofline plot to Appendix E.7 to illustrate the compute & memory access tradeoff.
>
> **Q: Evaluation on other memory technologies and accelerators**
> Our techniques are general, and we are excited about future work on applying these insights to other accelerators. Some accelerators have more SRAM than GPUs (e.g., TPUv4 with 128MB, Graphcore with 1GB compared to 19MB on A100), so we expect our method to be even more impactful.
>
> **Comparison with automatic fusion methods (e.g. TVM).**
> We discuss this in the common response. We have added comparisons with automatic fusion methods (Appendix E.5), where FlashAttention is still 2-3x faster. We are collaborating with compiler researchers to automate these techniques.

---

> > ### Comment · Reviewer_5Whx · 2022-08-05
> > **Rebuttal acknowledged**
> >
> > Thank you for the clarifications and for adding more data

---

### Official Review · Reviewer_ucdo · 2022-07-07

**Rating:** 7
**Confidence:** 4
**Soundness:** 4 excellent
**Presentation:** 4 excellent
**Contribution:** 3 good

**Summary:**

## Summary:
Problem: The paper addresses the problem of large memory requirements with high compute complexities of transformer models.
Solution: the paper attributes the above problem to the missing principle of IO-awareness, where the auditing is missing between different levels of GPU memory hierarchy. Which is a fair finding since the transformers are not designed with that in mind and moreover the GPUs at that time may not be having such complex memory hierarchies either or they may be evolving by then. The proposed solution is FLASHATTENTION
Results: The paper reports some exciting improvements to the speedup while also helping to improve the quality of models with the ability to process longer sequences.

## Distinction from state-of-the-art:
The paper clearly distinguishes from the state-of-the-art methods in the sub-domain and positions within the body of the knowledge.


**Questions:**

Please refer to the strengths and weaknesses above.

**Limitations:**

The paper describes the limitations to a satisfactory level.

**Strengths And Weaknesses:**

## Pros:
- The paper is written well, easy to understand and follow the concepts.

- The novelty of the paper is in identifying the underlying hardware causes of long time consumption by transformers and proposing the optimizations in the form of efficient data movement.

- The paper is a good fit for machine learning hardware software codesign in many respects.

## Cons:

- Line 54, typo, “out algorithm “ should be “our algorithm”

- One of the recent improvements are warp level matrix multiplications named as wmma in tensor core api. Since transformers involve a number of cascaded matrix multiplications starting from the input layer, wondering if you can compare the proposed approach with some of those mma/wmma based optimizations.

- I understand the focus of the paper is to perform efficient computations on GPU architectures. On the contrary, what if we  are operating on a resource constraint device (only in terms of memory but not that much on compute) where the SRAM size is not as big as that on the GPUs? Then, this approach looks to perform worse than the standard attention due to the N^2d^2. However, this can be out of the scope but can be listed as a limitation.

---

> ### Author Response · Authors · 2022-08-02
> **Response to Reviewer ucdo**
>
> Thank you for your comments.
>
> **Q: Comparison with warp-level matrix multiply and tensor cores.**
> The implementations we compare with (e.g., standard Pytorch implementation & Megatron-LM) all use warp-level matrix multiply in the matmul steps (Q @ K^T and Attention @ V). Our FlashAttention implementation also uses the warp-level matrix multiply for these steps. We benchmark with fp16 and bf16 precisions, where tensor cores are used for all methods. Overall FlashAttention is 2-4x faster than Pytorch and Megatron-LM implementations.
>
> **Q: Resource-constrained devices.**
> FlashAttention does require the device to have a certain amount of SRAM. We have tested with T4 GPU, which is marketed towards inference, and obtain 2-4.5X speedup (Figure 9). Other non-GPU accelerators often have more SRAMs than GPUs (e.g., TPUv4 has 128MB of SRAM and Graphcore has 1GB of SRAM compared to the 19MB of A100). An alternative to SRAM is to use register files instead to similarly reduce reads/writes to the device memory (e.g., GPU HBM). The general idea of being IO-aware is still important for speeding up deep learning operations.

---

> > ### Comment · Reviewer_ucdo · 2022-08-09
> > **Rebuttal acknowledged**
> >
> > Satisfied with the response of the authors and increasing the score to clear accept. Nice work.

---

### Official Review · Reviewer_uahY · 2022-07-10

**Rating:** 8
**Confidence:** 4
**Soundness:** 4 excellent
**Presentation:** 3 good
**Contribution:** 3 good

**Summary:**

This paper puts forward the hardware performance perspective in the design of attention algorithms. Specifically, it proposes to use established techniques such as tiling and re-computation to reduce the amount of high-bandwidth memory accesses for exact and approximate attention in transformers which is shown to be the main bottleneck in existing methods. Evaluation demonstrates that the proposed optimization enables training BERT and GPT-2 style architectures faster and achieves new state-of-the-art in the long range benchmark. Apart from training efficiency, the proposed attention improves performance in language modeling, long-document classification and it is the first result that is better than random in Path-X challenge.

**Questions:**

What is the runtime memory and decoding speed during training/inference for the models and baselines in Tables 1, 2, 4 and 5? An investigation beyond the runtime and memory of forward/backward pass of an individual attention layer is missing.

The introduction does not cover the complete view of prior work. Some emphasized the fact that intermediate results are often written into GPU memory and proposed unbiased approximations for iterative computation of attention e.g. kernel-based ones to avoid that and make attention compute-bound instead of memory-bound. Also, reducing the objective to reduce quadratic compute complexity is still important for scaling to extremely long sequences as this paper shows (e.g. Table 13) which is not reflected well in the text.

Do the authors plan to provide implementations for GPT-2 and BERT or the provided kernel would work?  Since it was highlighted that implementing a custom kernel is not straightforward this might be a concern reproducibility.

The algorithm appears to be applicable to pretrained models with non-optimized exact attention and does not necessarily require training the model from scratch. Is that the case and is there any evidence to support it?  If not please clarify this in the text.

For GPT-2 style models, could the authors explain how the causal masking is handled? It would be useful to include a description in section 3 for completeness and extra clarity.

**Limitations:**

Yes. Limitations require a bit more discussion (see comments above).

**Strengths And Weaknesses:**

**Strengths**
- In terms of quality, the exposition of ideas was clear and provided sufficient background for modern hardware which was necessary for the reader to follow. Writing overall was of high quality and the experimentation/proofs thorough.
- The paper brings awareness about reducing the number of memory accesses for computing exact attention which has been considered not scalable because of its quadratic complexity and memory access handling of existing implementations.
- The key idea of the algorithm is to compute attention sequentially by loading into on-chip memory blocks of Q, K, V and store the partial result along with easy-to-store statistics required to compute the final result into standard accelerator memory. This is based on the observation that modern accelerators have improved a lot in speed and access to standard memory can become a bottleneck for several standard operations used in deep learning (softmax, batch norm, etc). This kind of optimization is applicable to approximate attention as well and could help improve the efficiency of other operators in the future.
-  Experiments are well designed and provide evidence of the superiority of the proposed algorithm for both exact and approximate attention methods. It is quite important that this is the first attention that is exact and can extract information from such long contexts in high resolution and improves over all existing alternatives in tasks that require longer context. Should next generation of deep learning frameworks adopt such optimizations the impact can be big for scaling transformers to very long contexts without trading off performance.

**Weaknesses**
- Framing and motivation presents previous efforts as unaware of hardware performance and that they fail to speed up over existing transformers in terms of wall-clock time which is not universally true.
-  The main concern is how much hardware/deep learning framework/experiment-specific the low-level optimization results are. Also, the memory and speed numbers are thoroughly reported for attention layer but they are mostly missing for full models (except training speed).
- The code is provided which is helpful but implementations for big pretrained models such as BERT and GPT-2 are not available which might be an issue for reproducing results.

---

> ### Author Response · Authors · 2022-08-02
> **Response to Reviewer uahY**
>
> Thank you for your helpful feedback. We agree that FlashAttention could be impactful for scaling transformers to very long contexts without approximation. Indeed, several companies are looking to use FlashAttention to train language models with 8K context length or more.
>
> **Q: End-to-end memory reduction.**
> FlashAttention brings significant memory reduction: 10-20x if just counting the attention layer (Figure 3 in the paper) and 2-4x memory reduction for the full transformer model, depending on sequence length (1.8x for BERT-large, 4x for GPT2-small). It allows us to train with longer sequences and thus improves model quality. We have added these details to Section E.1 and E.2, thank you for this suggestion.
>
> **Q: Inference.**
> FlashAttention speeds up both the forward and backward passes, so it applies to inference as well. In our collaboration with the Pytorch developers, they have reported that FlashAttention helps their Pytorch implementation match the BERT inference latency of state-of-the-art inference systems in C++ (FasterTransformer with Tensor RT). We are also excited about future work on speeding up language generation (an iterative process where the query is often much shorter than the key and value), and we expect that IO-awareness (reducing memory reads/writes) will again be important for speedup.
>
> **Suggestions on contextualizing prior work.**
> We covered some of the efficient attention work (with sparsity and kernel approximation) in Appendix A. We will clarify the language in the introduction to reflect this and their contributions to addressing quadratic compute more directly.
>
> **Q: Code for BERT & GPT2**
> We have updated the supplemental file to include both the model code and the training script for BERT & GPT2, along with Docker files to reproduce the environment and our results. These will be made public after the review process. Thank you for this suggestion.
>
> **Q: Does FlashAttention apply to pretrained model.**
> Yes, since it simply computes the same attention as standard implementation (just faster), it applies to both models trained from scratch and pretrained models. In the “Long document classification” experiment in Sec 4.2, we directly replaced the attention module in a pretrained RoBERTa without modifying its weights, and finetuned it for the downstream tasks.
>
> **Q: Causal mask.**
> For the use case of autoregressive sequence modeling, FlashAttention with causal mask gives further speedup (around 1.8x) compared to FlashAttention without causal mask, as we only have to compute about half of the values of the attention matrix. This can be seen as a special case of block-sparse FlashAttention (Sec 3.3), where the zero blocks (the upper triangle) are skipped. We will add this description to Section 3.

---

> > ### Comment · Reviewer_uahY · 2022-08-06
> > **Rebuttal acknowledged**
> >
> > Thanks for the thoughtful replies and additional efforts, my major concerns have been addressed. To reflect that I increased my score.

---

### Official Review · Reviewer_UuW7 · 2022-07-11

**Rating:** 6
**Confidence:** 5
**Soundness:** 3 good
**Presentation:** 3 good
**Contribution:** 2 fair

**Summary:**

The paper leverages fusion and tiling techniques to reduce the overall memory consumption of attention mechanism. The results corroborate with some of the existing work on how fusion/tiling would help to reduce the memory footprint and runtime.

**Questions:**

(1) Fusion and tiling are well-established techniques in the literature, which is also highlighted in the paper. The block-level Softmax calculation is also not new. What are the main contributions of FlashAttention over existing methods?

(2) There are recent proposals to leverage fusing opportunities in transformers, such as [FLAT](https://arxiv.org/abs/2107.06419) and [Self-Attention Does Not Need O(n$^2$) Memory](https://arxiv.org/abs/2112.05682), how is FlashAttention different from these work?

(3) Which version of PyTorch did you use in your implementations? I am wondering if newer versions of PyTorch have already added some supports for fusion.

(4) I understand the limitation to study the scalability of the FlashAttention method over larger number of GPUs. However, I suspect that as we increase the number of GPUs the benefit of FlashAttention over existing methods may plateau. Do you have any intuition on the performance projection of FlashAttention as number of GPUs increase?

(5) What is the overhead of recompute step compared to the end-to-end training runtime?

(6) In Figure 2-Middle, I suspect if you include the data for larger block sizes the number of HBM accesses would increase due to the limited on-chip resources. Is my understanding correct?

(7) In addition, in the most recent release of [MLPerf](https://mlcommons.org/en/training-normal-20/), the training runtime has been improved across all the models. I understand that the NeurIPS submission deadline is before the most recent MLPerf release, but I am wondering if you could provide some insights on whether FlashAttention would outperform the SOTA implementations?

(8) This is more a general comment about transformer models and the contribution of each layer to the overall runtime. Even for reasonably large sequence lengths, FF layers are still the main contributor to the end-to-end runtime. I am wondering if we are targeting the right problem to reduce the runtime of transformer models (I understand that memory consumption is still a large part of self-attention layers, but I doubt that this statement is correct for their runtime).

**Minor**

- There is a missing reference in p. 23.
- In some of the tables, the accuracy of FlashAttention is lower than SOTA. Nothing major, but would be good to make sure that in all cases, the final accuracy matches the SOTA implementation. This would provide a better head-to-head comparison.

**Ethics Review Area:**

["I don’t know"]

**Limitations:**

The authors summarizes some of the main limitations of this work in a dedicates section. As authors clearly mentioned, this is not a one-fit-all solutions for all the platforms and models. In this current state, the engineering efforts to adopt such approach may not be transferrable to other platforms and ML libraries.

**Strengths And Weaknesses:**

**Strengths**

- The paper consider both forward pass and backward pass in the results that makes it more interesting compared to recent fusion/tiling approaches.

- The paper compares with the hand-tuned MLPerf results (although it is old by now) and show up to 14% improvement for one transformer model.

**Weaknesses**

- Fusion/tiling is not new and has been extensively explored in the literature. It is not clear what the contributions of this work are over previous methods.

- The paper seems to merely provide an engineering solution, merging multiple known approaches (fusion, tiling, algebraic aggregation) --- some already applied to transformer models.

- The paper needs to perform head-to-head comparison with existing fusion techniques for transformers.

---

> ### Author Response · Authors · 2022-08-02
> **Response to Reviewer UuW7**
>
> Thank you for your constructive comments and insightful questions.
>
> **Q: Head-to-head comparison with existing fusion techniques for transformers.**
> We discussed this in the common response. We have added comparisons with automatic fusion methods (Appendix E.5), where FlashAttention is still 2-3x faster. Thank you for this great suggestion.
>
> **Q1: Main contribution and comparison with other fusion techniques:** we discuss this in the common response.
>
> **Q2: Comparison with Rabe & Staats algorithm.**
> We have updated the paper with a discussion of the differences (Appendix B.5). While both algorithms offer memory saving, our algorithm focuses on reducing memory IOs (not just total memory requirement), and thus yields substantial speedup. Rabe & Staats, on the other hand, focuses on reducing the total memory requirement and is slower than or on-par with standard implementation in the forward pass. Moreover, while Rabe & Staats relies on generic gradient checkpointing, FlashAttention simplifies the backward pass analytically (Appendix B.2 & B.4), which again yields speedup in the backward pass and saves even more memory.
>
> **Q2: Comparison with FLAT.**
> FLAT proposes an attention fusion technique for custom dataflow accelerators, while FlashAttention works with commonly used GPUs. FLAT lacks the softmax decomposition technique (line 151), and thus needs to compute softmax over an entire row (or several rows) of the attention matrix. Therefore, it requires custom hardware with large SRAM to fit the entire key sequence, which is not yet practical for GPUs.
>
> FLAT’s speedup is measured with a hardware simulator. On the other hand, FlashAttention yields wallclock speedup and memory saving on commodity GPUs by operating on blocks (requiring relatively much smaller SRAM compared to custom accelerators). As mentioned in the “techniques” paragraph in our common response, this is thanks to both the softmax decomposition and operator fusion techniques.
>
> **Q3: Pytorch version:** We have tested on the latest Pytorch version (both 1.12 and nightly), and they do support fusing softmax and masking (but not other operations in attention). We added this comparison in Appendix E.5. One subtlety that makes automatic fusion difficult is that the softmax operation needs to be decomposed algebraically before fusion could be applied. We hope that advances in compilers will enable these speedup / fusion in future versions.
>
> **Q4: Performance with more GPUs.**
> In our benchmarks, FlashAttention yields consistent speedup when scaled from 1 GPU to 8 GPUs (in fact, most of our end-to-end speedup reported in Sec 4 are on 8 GPUs). We have not benchmarked the scenario with >8 GPUs due to lack of available hardware. On more GPUs across machines and nodes, the communication time may become more important, but several techniques exist to overlap computation and inter-GPU communication (e.g., pipeline parallelism). If there are other reasons that speeding up attention will have less benefit on >8 GPUs, we would love to understand them.
>
> **Q5: Overhead of recomputation.**
> Recomputation is fast, since the inputs are already in SRAM (attention is bottlenecked by memory reads/writes), and it is done as part of the backward kernel. Figure 2 Left shows a comparison with Pytorch attention implementation where FlashAttention incurs more FLOPs (13% more) due to recomputation but reduces IOs by 9.1X, leading to 5.7X speedup.
>
> **Q6: What happens when one increases the block size too much.**
> In Figure 2 Middle, we show block size up to 512. Larger block sizes simply do not fit into the available SRAM on an A100 GPU. Your understanding is right in that block size 512 does about the same as block size 256, since other resources (e.g., compute) become the bottleneck.

---

> > ### Author Response · Authors · 2022-08-02
> > **Response to Reviewer UuW7 - part 2**
> >
> > **Q7: Comparison with MLPerf implementation of BERT:** We compare against the newest implementation from Nvidia at the time of submission (MLPerf 1.1 in December of 2021), and it was the fastest implementation of BERT that we could find. For context, this implementation is about 2.8x faster than the widely-used Huggingface BERT (Appendix E.1). We have added this citation (Dec 2021) to the MLPerf citation (2020) to make this clear.
> >
> > After the submission deadline, new MLPerf submissions (MLPerf 2.0, June 2022) from many hardware vendors have made exciting progress in both software optimization and hardware tuning (e.g. GPU overclocking, liquid cooling). We are happy to report that our MLPerf BERT submission (running on a publicly available Azure cloud instance) is the fastest single-node submission on cloud instances. Across all single-node submissions (cloud & on-premise machines), only two submissions (e.g., with GPU overclocking & software optimizations tailored to MLPerf benchmark) are faster.
> >
> > Our general method surprisingly is still faster than most submissions from vendors (e.g., Nvidia), which are often the result of teams of hardware and software engineers working for 6 months (duration between MLPerf rounds). This is very encouraging to us: simple ideas can be competitive with (or even beat) solutions from ML performance experts at these large companies.
> >
> > **Q8: Attention vs FFN as the bottleneck of Transformers:** for the large language models currently in use (e.g., >1B) the FFN layers take more time than attention, as the hidden dimension is often much larger than the context length, but future models could have much longer context length. We speculate that one of the reasons these models are scaled by increasing hidden dimension (and not context length) is that the existing attention implementation is slow and memory-hungry on long sequences. One would then have to use larger hidden dimensions so that the FFN layers take more time proportionally (compared to attention), in order to make efficient use of hardware. Our work is a step toward changing this calculus: beyond making models wider and deeper, we can also increase their context length (Sec 4.2). With attention being faster and more memory-efficient, one exciting direction is to train large language models with longer context. We are happy to report that several companies are looking to use FlashAttention to train language models with context length 8k or more.
> >
> > **Q: Matching the accuracy/perplexity of reference models.**
> > We have fixed a small bug with our training setup and rerun the experiments to verify that GPT2-medium FlashAttention yields the same perplexity as the reference model from Huggingface (Table 2). We previously put weight decay on the LayerNorm parameters of the FlashAttention runs by accident, causing a 0.1 perplexity difference. Fixing that yields the same perplexity between the two models.

---

> > ### Comment · Reviewer_UuW7 · 2022-08-08
> > **Rebuttal Discussion**
> >
> > Thanks for providing the additional explanations.
> >
> > **Q1: Head-to-head comparison with existing fusion techniques for transformers**
> >
> > How did you choose this batch size? Figure 1 uses batch size of 64 while for this study you use a batch size of 32.  It is confusing that different batch sizes are used across the paper. Which brings me to a follow-up question, how sensitive FlashAttention is with regards to the batch size?
> >
> > **Q2: Comparison with FLAT**
> >
> > I doubt that the FLAT implementation is specific to a custom accelerator, however I agree with the authors that FLAT's minimum granularity is an entire row.
> >
> > **Q4: Performance with more GPUs.**
> >
> > As we increase the number of GPUs, the opportunity for overlapping the communication and computation could potentially decrease (or plateau) as now each GPU spends less time on computation. One experiment that could be interesting to obtain is to see how the communication/communication overhead changes as you increase the number of GPUs.
> >
> > Thanks for fixing the typos and additional bugs in the code and providing more context about the experiments.

---

> > > ### Author Response · Authors · 2022-08-09
> > > **Followup response to Reviewer UuW7**
> > >
> > > Thank you for the productive discussion.
> > >
> > > **Q1: Comparison with fusion techniques and is FlashAttention sensitive to batch size?**
> > > FlashAttention is not particularly sensitive to batch size. We updated the benchmark with automatic fusion (Table 11) to include batch sizes 16, 32, 64. Across all these batch sizes, FlashAttention is 2-3x faster than automatic fusion methods and Megatron-LM.
> > >
> > > The difference between Figure 2 and Table 11 was due to dropout vs no dropout, and key-padding mask vs causal mask. We have updated Figure 2 to be consistent with Table 11 (same batch size, key-padding mask, no dropout). Thank you for this great suggestion.
> > >
> > > **Q2: Comparison with FLAT.**
> > > To the best of our understanding, FLAT does not run on GPUs. On the other hand, our algorithm (e.g., with softmax decomposition) allows FlashAttention to work with commodity GPUs. As an example, FLAT evaluates on hardware simulators, varying the number of processing elements (PE) and the on-chip & off-chip bandwidth (Table 2 of FLAT paper). FlashAttention evaluates on GPUs commonly used to train Transformers models (e.g., T4, A100, RTX 3090) and achieves wallclock time speedup. We agree that co-designing hardware and software/algorithms is an exciting direction to further speed up model training.
> > >
> > > **Q4: Performance with more GPUs.**
> > > Speeding up any part of the model (e.g., attention, MLP, LayerNorm) will have less of an effect if communication overhead becomes significant. Fortunately, in the case of training large models on tens to hundreds of GPUs, several techniques (data/tensor/pipeline parallelism) have been developed to minimize communication overhead. As an example, Megatron-LM reports that training larger models on more GPUs increases the efficiency per GPUs (Table 1 of https://developer.nvidia.com/blog/scaling-language-model-training-to-a-trillion-parameters-using-megatron/).
> > > Thanks to your suggestion, we have added an experiment (Table 8) measuring the speedup on GPT-2 small as we increase the number of GPUs from 1 to 8 (using the hardware we have available). The speedup is between 3.5x and 3.7x.
> > > We are excited about future work on applying IO-aware techniques to distributed training.

---

### Official Review · Reviewer_jjqW · 2022-07-12

**Rating:** 8
**Confidence:** 3
**Soundness:** 4 excellent
**Presentation:** 4 excellent
**Contribution:** 4 excellent

**Summary:**

This paper studies an important topic for current Transformer architectures: accelerating the speed and memory consumption for self attention operations. The proposed method leverages tiling and recomputation to reduce the memory access of HBM, avoiding the memory bound issues in standard self attention kernels. The authors also extend the proposed Fast-Attention with block-wise sparsity, and  is the fastest implementation among exsiting approximate attention methods. The experiments show the superior performance w.r.t. memory and speed against existing solutions. The paper is quite solid with technical details, yet it is still well-written and friendly to the general audience. The idea to exploit IO awareness should be inspiring for the community to develop more efficient algorithms and kernels.

**Questions:**


- What are the separative gains of reduced HBM access and tiling? In Algorithm 0, the Q,K,V matrices are loaded to HBM by blocks. I wonder whether similar techniques (tiling) are applied in standard attention implementation?

- The baselines in Table 2 and 3 do not fuse the attention kernel. I wonder if there are any public baselines with fused kernels for fair comparisons?

- As a suggestion, it is a bit miss-leading to show that $\Theta(N^2 d^2 M^{-1})$ requires less memory access than $\Theta(Nd + N^2)$, which may be attributed to tiling (due to the presence of $M^{-1}$). The main reduction of HMB access should come from not storing the attention score $S$, which takes $\Theta(N^2)$ accesses.


**Ethics Review Area:**

["I don’t know"]

**Limitations:**

The authors have listed and discussed the potention directions for the limitations.

**Strengths And Weaknesses:**

- The paper studies an important research question: designing fast and memory-cheap attention kernels for the general transformer architectures. The authors conduct extensive experiments and achieve impressive results w.r.t. training time and memory consumption across multiple tasks and network architectures.

- The proposed Fast-Attention can be readily combined with block-sparsity to allow longer sequences, and the results are still faster than existing approximate attention designs.

- The writing is clear and self-contained, rigorous and yet friendly to the general audience. Though I am not an expert in low-level designs on CUDA but I find it rather pleasing to read the aritcle.

---

> ### Author Response · Authors · 2022-08-02
> **Response to Reviewer jjqW**
>
> Thank you for the helpful feedback.
>
> **Q: What are the separate gains of reduced HBM access and tiling?**
> One needs tiling (and softmax decomposition, cf. the paragraph on techniques in our general response) to reduce the HBM access. Tiling alone (without fusion to reduce HBM access) can save memory but generally slows down attention. An example is the attention algorithm of Rabe & Staats [69], which offers substantial memory saving but has around 40% slowdown compared to standard implementation.
>
> **Q: Comparison with fused attention implementation.**
> We discuss this in the common response. For GPT2 we compared against the implementation from Megatron-LM, which fuses the softmax and masking step. We have added comparisons with automatic fusion methods (Appendix E.5), where FlashAttention is still 2-3x faster.
>
> **Suggestion on theoretical gains from tiling vs HBM access.**
> We see tiling as a necessary step to reduce HBM access, at least in the case of FlashAttention. We will clarify this in the text. Thank you for this suggestion.

---

> > ### Comment · Reviewer_jjqW · 2022-08-08
> > **Response to authors**
> >
> > Thanks for the clarification by the authors.

---

### Author Response · Authors · 2022-08-02
**Common Response [1/2]**

We thank the reviewers for insightful comments and constructive feedback. We are happy that all reviews were positive, and that reviewers thought that our work addresses an important problem with well-designed experiments, and that our paper was clear and well-written.

We first report some updates, address a few common questions, and then respond to specific questions from the reviewers.

We are happy to report several updates to FlashAttention since submission. We are happy to see that FlashAttention has already begun making an impact in a short time:
- **Getting FlashAttention into people’s hands:** we are working with the PyTorch developers to integrate FlashAttention into torch.nn.Transformer, potentially benefiting a much larger audience of researchers and practitioners in the near future. FlashAttention has also been independently reimplemented in Triton (a high-level language embedded in Python) and Jax, allowing for easier experimentation by other researchers.

- **Latest MLPerf benchmark result:** Our MLPerf submission (Sec 4.1) has been officially verified by MLCommons (MLPerf organizers), and our submission is the fastest single-node BERT model running on cloud instances. This verification is a month-long manual review to ensure our submission follows all the MLPerf training rules and compares fairly with other submissions (e.g, exactly the same BERT model, training data, eval data, starting from the same checkpoint and reaching the same validation accuracy).

- **Collaborations on modeling longer sequences:** We have seen exciting progress in modeling longer sequences with FlashAttention, which is our original motivation. Several major Internet companies and startups are looking to use FlashAttention to train large language models with context length 8K or more (currently large language models such as GPT3 are limited to 2K context length) and to train Vision Transformers on high-resolution images.

We now address some common questions about our contributions, techniques, and baselines.

**Contribution: a conceptually simple approach making progress on a well-studied problem - speeding up attention.** Several reviewers point out that the high-level techniques we used are known (as we made clear in the paper, lines 49 and 145). If this were a new problem, a simple answer using well-known techniques might suggest that the problem is not interesting or that the solution is obvious. However, speeding up attention is arguably one of the most studied and economically valuable problems in machine learning in the past 5 years, with huge teams of researchers and engineers working on it (from major deep learning frameworks, many large companies, and academia, cf. paragraph below on baselines). Yet FlashAttention can still speed it up substantially with no approximation (2-4x speedup, 10-20x memory saving for the attention layer). That we showed a classic viewpoint was even competitive, and in fact missing from the state of the art, is to our mind elegant and important. Moreover, this “simple technique” of being IO-aware was at the core of large scale data processing (e.g., relational databases), and its implications played out over several generations. Being IO-aware required massive redesign to databases and data management systems, so it’s a big idea and viewpoint that we look forward to applying to more areas.

**Techniques: one needs both softmax decomposition (tiling) and operator fusion.** It was surprising to us that even though all the high-level techniques (tiling & recomputation) are available, there was still so much headroom (2-4x) in speeding up exact attention. It might not have been obvious how to apply tiling in the presence of softmax (line 47) and whether recomputation (which incurs more FLOPs) could yield speedup (line 52). We speculate two reasons why such substantial improvement is possible.

(1) The technique of softmax decomposition with scaling (line 151) is known to many ML algorithm researchers but might not have been obvious to many systems researchers, while operation fusion / memory IOs reduction is the bread-and-butter of the systems / compilers community but is not as familiar to algorithm researchers. In our case of FlashAttention, one needs both the softmax decomposition and the operation fusion to achieve speedup and memory saving.

(2) As mentioned in our introduction (line 29), there has been a focus on attention approximation that aims to reduce FLOPs. While these methods bring novel algorithmic ideas, many do not yield wallclock-time speedup (attention is bottlenecked by memory reads/writes and not compute), and they all have to trade off model quality. FlashAttention speeds up attention in wall-clock time, while maintaining the same model quality (when we do not change the model, Sec 4.1) or improving model quality (by training on longer sequences, Sec 4.2).

---

> ### Author Response · Authors · 2022-08-02
> **Common Responds [2/2]**
>
> **Baselines: we compared against the strongest baselines we could find**, and FlashAttention brings substantial end-to-end speedup (15% for BERT, 2.0-3.5X for GPT2) and memory saving (2-3x) even against these strong baselines. For BERT, it’s the MLPerf implementation from Nvidia that set the fastest training record at the time of submission, which also fuses all attention steps. This implementation is written specifically for BERT-large in the MLPerf benchmark (seqlen at most 512, head dim 64 only, A100 only), and is 2.8x faster than Huggingface BERT (Appendix E.1). MLPerf implementations from vendors are often the result of teams of hardware and software engineers working for 6 months (duration between MLPerf rounds). For GPT2, one of our baselines is the implementation from Megatron-LM (also from Nvidia), which fuses the softmax and masking steps. The Megatron-LM implementation of Transformers has been used to train some of the largest language models (Megatron-Turing NLG, OPT, and BLOOM).
>
> Thanks to the reviewers’ suggestions, we have added comparison against automatic fusion baselines in Appendix E.5 (NVfuser in the newest version 1.12 of PyTorch, AOT compiler from functorch, and TVM). Some of these automatic fusion methods are able to fuse softmax + masking (but not other steps), but they are still slower than the Megatron implementation. FlashAttention is still 2-3x faster than these methods. One reason is that fusion can only be fully exploited once softmax is analytically decomposed into local softmaxes (line 151, cf. paragraph above on techniques). We hope that advances in compilers will enable these speedup / fusion in the future. In fact, we are collaborating with compiler researchers to automate these techniques.
>
> **Outlook:** FlashAttention is an example where exciting opportunities open up at the intersection of hardware & algorithms. We hope that our work inspires future research in this intersection.

---

### Meta-Review · Area_Chair_f4KQ · 2022-08-30

**Recommendation:** Accept
**Confidence:** Certain

**Metareview:**

The authors study the problem of improving wall-clock time of performing exact and approximate attention in Transformer networks. Towards this, they identified the number of HBM access as a limiting factor. The authors propose FlashAttention which employs tiling and fusion operations to efficiently implement the attention operation on (GPU) SRAM while minimizing the number of HBM accesses. Even though the proposed approach has higher FLOP coun wrt. the standard attention implementation, it leads to reduced wall clock time due to the smaller number of HBM accesses. The proposed FlashAttention can be easily modified to obtain a sparse attention method, namely block-sparse FlashAttention. The paper is very well written and the authors demonstrate the utility of the proposed FlashAttention/block-sparse FlashAttenion on a wide range of settings. On many benchmarks, the proposed method allows for a longer context length, which leads to higher performance as compared to the existing approaches in the literature.

Overall, the findings of the paper are very interesting and impactful. All the reviewers are quite positive about the paper. During the rebuttal phase the authors have included additional experiments that further strengthen the contributions and value of the paper. Some of the reviewers pointed out that the paper builds on well-known techniques like tiling and fusion. However, given that the paper makes significant improvements on a timely and actively studied problem of making the attention operation efficient and the results in the paper have the potential to inspire many follow-up explorations, I recommend that the paper be accepted in NeurIPS 2022.

**Award:**

No

---

### Decision · Program_Chairs · 2022-09-14

Accept